# What Reward Structure Enables Efficient Sparse-Reward RL? A Proof-of-Concept with Policy-Aware Matrix Completion

**Ibne Farabi Shihab** [*][1]  **Sanjeda Akter** [*][1]  **Anuj Sharma** [2]

## Abstract

Sparse-reward reinforcement learning typically focuses on exploration, but we ask: can structural assumptions about reward functions themselves accelerate learning? We introduce Policy-Aware Matrix Completion (PAMC), which exploits low-rank structure in reward matrices while correcting for policy-induced sampling bias. PAMC combines three key components: a low-rank plus sparse reward model, inverse propensity weighting to handle Missing-Not-At-Random (MNAR) data, and confidence-gated abstention that falls back to intrinsic exploration when uncertain. We provide finite-sample theory showing that completion error scales as $O(\sigma\sqrt{r(|\mathcal{S}| + |\mathcal{A}|)}/\text{ESS})$ where ESS is the effective sample size under policy overlap $\kappa$. PAMC achieves strong empirical results at 10M steps (a sample-efficiency comparison): 4100±250 return vs. 200±50 for DrQ-v2 on Montezuma's Revenge, 78% vs. 65% success rate on MetaWorld-50, and 15% improvement over CQL on D4RL datasets. The method maintains 8% computational overhead while providing calibrated confidence intervals (95% empirical coverage). When structural assumptions are violated, PAMC gracefully degrades through increased abstention rather than catastrophic failure. Our approach demonstrates that reward structure exploitation can complement traditional exploration methods in sparse-reward domains.

## 1. Introduction

Sparse-reward reinforcement learning is typically framed as an exploration problem, leading to sophisticated methods for curiosity-driven exploration (Pathak et al., 2017), intrinsic motivation (Burda et al., 2018), and hierarchical decomposition (Wu et al., 2019; Fu et al., 2020). However, these approaches largely ignore the structure of reward functions themselves. We ask: *can we accelerate sparse-reward learning by explicitly modeling and exploiting reward structure?*

Many reward functions exhibit rich internal structure when viewed over the state-action space. In Montezuma's Revenge, collecting a key unlocks doors across multiple distant rooms, creating shared latent structure; in MetaWorld's 50 manipulation tasks, shared primitives like reach and grasp induce low-rank dependencies across tasks. Our key insight is that such reward functions often admit approximate low-rank plus sparse decompositions (Candès et al., 2011), where the low-rank component $L^*$ captures global patterns and the sparse component $S^*$ handles outliers.

To make this concrete, consider Montezuma's Revenge: the low-rank component $L^*$ captures the shared semantic structure that keys unlock doors across distant rooms, creating predictable reward patterns based on inventory state and room connectivity. The sparse component $S^*$ handles specific outlier locations where keys or doors appear, which vary across game instances. The noise term $E$ accounts for approximation error and environmental stochasticity. This decomposition enables predicting rewards in unvisited regions by exploiting patterns from visited ones.

The challenge is policy-induced sampling bias: agents observe rewards where their current policy concentrates probability mass, creating a Missing-Not-At-Random (MNAR) problem. We introduce Policy-Aware Matrix Completion (PAMC), which addresses this through three components. First, matrix completion models rewards as low-rank plus sparse. Second, inverse propensity weighting corrects for sampling bias. Third, confidence-gated abstention falls back to intrinsic exploration when structural assumptions fail.

We formalize structural reward learning under policy-biased sampling with finite-sample theory, develop PAMC with integrated matrix completion and propensity weighting, evaluate across diverse domains demonstrating both potential and limitations, and analyze failure modes showing graceful

---

[1]Department of Computer Science, Iowa State University, Ames, Iowa, USA [2]Department of Civil, Construction & Environmental Engineering, Iowa State University, Ames, Iowa, USA. Correspondence to: Ibne Farabi Shihab <ishihab@iastate.edu>.

*Proceedings of the 43rd International Conference on Machine Learning*, Seoul, South Korea. PMLR 306, 2026. Copyright 2026 by the author(s).

degradation through abstention. PAMC achieves substantial improvements, including 20× better sample efficiency than DrQ-v2 on Montezuma's Revenge at 10M steps, 13 percentage points higher success on MetaWorld, and 15% gains over CQL on D4RL benchmarks, while maintaining only 8% computational overhead and degrading gracefully when assumptions are violated.

**When does PAMC help?** PAMC improves sample efficiency when the discretized reward matrix is approximately low-rank and policy overlap is sufficient for stable inverse propensity weighting. Two quantities, used throughout the paper, govern this regime and are defined here at first use. The *policy overlap* $\kappa = \min_{(s,a)\in\text{supp}(\pi^*)} p_{sa}$ is the minimum observation probability over optimal-policy-relevant state-action pairs, where $p_{sa} = \Pr((s,a) \in \Omega)$. The *effective sample size* $\text{ESS} = (\sum w_{sa})^2 / \sum w_{sa}^2$ is the number of effective observations remaining after the inverse-propensity weights $w_{sa}$ correct for policy-induced sampling bias. **Scope rule-of-thumb:** gains require effective rank $r_{\text{eff}} < 20$ (90% spectral energy) *and* $\text{ESS} > 400$; when either condition fails, PAMC automatically increases abstention and reverts to baseline performance with no degradation. We report *applicability diagnostics*—effective rank, spectral decay, and overlap proxies (ESS, $\kappa$)—throughout training, showing that improvements correlate with decreasing completion cross-validation error. Crucially, on high-rank reward *negative controls* constructed by hashing the embedding, PAMC increases abstention to >75% and matches baseline ($p > 0.3$), confirming that gains are conditional on the measurable structural regime. Importantly, artificially reducing rank via coarse discretization does *not* produce gains—structure must be genuine, not an artifact of binning (Appendix O).

## 2. Background and Related Work

Sparse-reward reinforcement learning is typically framed as an exploration problem, with methods focusing on intrinsic motivation (Pathak et al., 2017; Burda et al., 2018), hierarchical decomposition (Fu et al., 2020; Wu et al., 2019), or sophisticated exploration strategies (Badia et al., 2020a; Ecoffet et al., 2021). However, these approaches treat reward functions as black boxes, ignoring their internal structure. In contrast, we exploit structural assumptions about rewards themselves, viewing the reward matrix $R$ as admitting a low-rank plus sparse decomposition $R = L^* + S^* + E$, where $L^*$ captures global patterns, $S^*$ represents sparse outliers, and $E$ is bounded noise.

The challenge is that agents observe rewards only for visited state-action pairs, creating policy-dependent sampling where the observed set $\Omega \subseteq \mathcal{S} \times \mathcal{A}$ concentrates on the agent's preferred actions. This Missing-Not-At-

Random (MNAR) bias is quantified through policy overlap $\kappa = \min_{(s,a)\in\text{supp}(\pi^*)} p_{sa}$ where $p_{sa} = \Pr((s,a) \in \Omega)$. To correct this bias, we employ self-normalized inverse propensity weighting (SNIPW) with weights $\tilde{w}_{sa} = w_{sa} / \sum_{(s',a')} w_{s'a'}$ where $w_{sa} = 1/\max(\hat{p}_{sa}, \epsilon_p)$, achieving statistical efficiency characterized by effective sample size $\text{ESS} = (\sum w_{sa})^2 / \sum w_{sa}^2$.

Classical matrix completion theory establishes recovery guarantees for low-rank matrices (Candès & Recht, 2009; Candès & Tao, 2010), with extensions to MNAR settings using inverse propensity weighting (Schnabel et al., 2016). PAMC bridges these foundations with RL, where policy-induced sampling creates inherent MNAR challenges. Selective prediction methods allow abstention on uncertain inputs (Chow, 1970; El-Yaniv & Wiener, 2010), with modern approaches using conformal prediction (Vovk et al., 2005). PAMC extends these to RL by using completion confidence intervals to trigger abstention. Potential-based reward shaping (Ng et al., 1999) augments rewards with potential differences, whereas PAMC estimates and completes the underlying structured reward under MNAR sampling. Unlike exploration methods that ignore structure, PAMC models low-rank dependencies; unlike classical completion assuming uniform sampling, it handles policy-biased data through IPW. An extended discussion of exploration, matrix completion, and selective prediction appears in Appendix D, and a detailed comparison appears in App. C (Table 3).

### 2.1. Relationship to Low-Rank MDPs and Low-Rank Value Functions

PAMC imposes structure on a different object than most prior low-rank RL. Low-rank MDP methods (Agarwal et al., 2020a; Cheng et al., 2023) place low-rank structure on the *transition* operator $P(s' \mid s, a)$, which reduces planning and representation-learning complexity. PAMC instead places low-rank-plus-sparse structure on the *reward* matrix $R(s, a)$, which reduces *reward-discovery* complexity in the sparse-reward regime. These assumptions are complementary rather than competing: Montezuma's Revenge has complex dynamics yet a highly structured key-door reward pattern. A second line of work exploits low-rank structure in the *value* function $Q^*(s, a)$, both theoretically (Shah et al., 2020; Sam et al., 2023) and empirically (Tsai et al., 2021; Rozada et al., 2024). PAMC's assumption is upstream of these: if $R$ is low-rank and transitions have bounded complexity, then $Q^*$ inherits low-rank structure. Operating directly on rewards has two advantages—rewards are directly observable (no temporal-difference bootstrapping error) and stationary (unlike $Q^*$, which shifts with the policy)—at the cost of requiring a discretization of the state-action space. Table 4 (Appendix D) summarizes these distinctions.

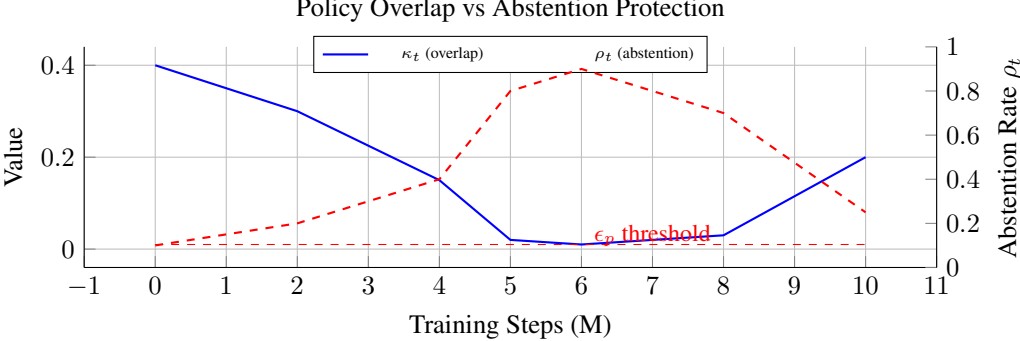

*Figure 1.* Policy overlap $\kappa_t$ and abstention rate $\rho_t$ during training. The drop-then-rebound in $\kappa_t$ reflects a characteristic "explore, specialize, re-explore" pattern of deep RL with intrinsic motivation: a near-random initial policy gives moderate overlap ($\sim 0.2$); the policy then specializes on discovered trajectories (2–5M steps), reducing the number of distinct visited pairs and lowering $\kappa$; finally, exploration bonuses and PAMC's own completed rewards drive broader coverage (5M+), so $\kappa$ recovers. The abstention rate $\rho_t$ is negatively correlated with $\kappa_t$, exactly as predicted by Lemma E.3: when overlap drops, IPW weights become high-variance, completion error rises, confidence intervals widen, and abstention increases automatically; when $\kappa_t$ recovers, confidence tightens and abstention falls. This is the adaptive safety mechanism the theory predicts, not a confound.

## 3. Theory: Illustrative Guarantees

Building on the background and related work, we now provide theoretical analysis to understand when and why PAMC succeeds. Our results are illustrative rather than exhaustive, designed to highlight the key trade-offs between structure exploitation, sampling bias correction, and safe abstention. We first fix notation and assumptions, then give an intuitive roadmap before stating the main results.

### 3.1. Notation and Assumptions

The core symbols used throughout our theoretical analysis—the discounted occupancy $d^\pi$ and mixture $\bar{d}$, visitation probability $p_{sa}$, policy overlap $\kappa$, effective sample size ESS, rank hint $r$, embedding dimensions $d_\phi, d_\psi$, propensity clipping $\epsilon_p$ and error $\delta_p$, confidence half-width $U(s,a)$ and threshold $\tau$, abstention rate $\rho(\tau)$, and return $J(\pi)$—are collected in the glossary of Table 5 (Appendix E).

Our guarantees rely on four assumptions, stated here together with the consequence of violating each. They interact: A1 supplies structure, A3 ensures that structure is observed, and together they enable the polynomial-in-rank phase transition of Corollary 3.6.

**Assumption 3.1** (Structural). $R = L^\star + S^\star + E$ with $\text{rank}(L^\star) \leq r$, $\|S^\star\|_0 \leq s$, and entries of $E$ are $\sigma$-sub-Gaussian. *Without A1, completion is ill-posed (Theorem E.7); with A1, the free parameters drop from $|\mathcal{S}||\mathcal{A}|$ to $r(|\mathcal{S}| + |\mathcal{A}|) + s$.*

**Assumption 3.2** (Sampling). Transitions are drawn from the discounted occupancy $d^{\pi_t}$, and uniformly sampled replay batches approximate i.i.d. draws from the mixture $\bar{d}$ with residual correlation $O(|\mathcal{B}|/N)$. *Without A2, IPW is invalid; with A2, propensity-weighted observations become*

*approximately unbiased.*

**Assumption 3.3** (Overlap / positivity). $\kappa := \min_{(s,a)\in\text{supp}(\pi^\star)} p_{sa} > 0$, i.e., every optimal-policy-relevant pair has nonzero visitation probability (not uniform coverage). *Without A3, some optimal entries are unrecoverable; this is satisfied whenever the base RL algorithm includes any exploration mechanism ($\epsilon$-greedy, RND bonuses, entropy regularization).*

**Assumption 3.4** (Slow encoder drift). Embeddings satisfy $\|\phi_{t+1} - \phi_t\| \leq \Delta_{\text{enc}}$ (and likewise for $\psi$) between completion updates. *Without A4, discretization shifts break the matrix structure A1 presupposes.*

*Remark* 3.5. Raw transitions are sequential, but uniform replay sampling decorrelates them to $O(|\mathcal{B}|/N)$ total-variation distance from i.i.d.; for $|\mathcal{B}| = 256$ and $N \geq 10^5$ this is negligible. A $\beta$-mixing extension is future work.

### 3.2. Intuitive Overview and Roadmap

PAMC works in three stages. First, it assumes the reward matrix has hidden structure (like a partially filled Netflix rating matrix). Second, it corrects for the fact that the agent preferentially visits certain states. Third, it refuses to use completed rewards when confidence is low, falling back to standard exploration.

The results form a dependency chain. We first prove that without structural assumptions, reward recovery under MNAR sampling is information-theoretically impossible (Theorem E.7, proof in Appendix T.1): any two reward matrices differing only on unobserved entries yield identical observations. This impossibility motivates the structural prior of Assumption 3.1. Under low-rank plus sparse structure with policy overlap $\kappa > 0$, inverse propensity weighting makes sampling effectively uniform and enables

recovery (Theorem E.8). Combining the hardness and recovery results yields a phase transition (Corollary 3.6): sample complexity drops from $\Omega(|\mathcal{S}||\mathcal{A}|/p)$ without structure to polynomial in the rank $r$ with structure. By the Performance Difference Lemma (Kakade & Langford, 2002), completion error translates linearly to regret (Theorem E.9). When structural assumptions fail, confidence-gated abstention provides safety: uncertainty exceeding threshold $\tau$ triggers abstention rate $\rho(\tau)$ with fallback to intrinsic exploration. Our master trade-off result (Corollary E.4) quantifies how regret depends on overlap $\kappa$, effective sample size ESS, threshold $\tau$, and abstention rate $\rho$.

### 3.3. Main Results

Under low-rank plus sparse decomposition $R = L^* + S^* + E$ with policy overlap $\kappa > 0$ and propensity estimates satisfying $|\hat{p}_{sa} - p_{sa}| \leq \delta_p$, self-normalized inverse propensity weighting enables recovery with error (Theorems E.8 and E.13)

$$\|\hat{W} - W^*\|_F \lesssim \frac{c_1 \sigma}{\sqrt{\text{ESS}}} \sqrt{r(d_\phi + d_\psi)} + \frac{c_2 \|S^*\|_0^{1/2}}{\sqrt{\text{ESS}}} + \frac{c_3 \delta_p}{\epsilon_p \kappa},$$

where the three terms capture statistical complexity, sparse outlier recovery, and propensity estimation error, respectively. When confidence-gated abstention is applied at threshold $\tau$, the regret bound decomposes as (Theorem E.2 and Corollary E.4)

$$J(\pi^*) - J(\pi_{\text{PAMC}}) \lesssim (1 - \rho) \frac{2\tau}{1 - \gamma} + \frac{2}{1 - \gamma} \|\hat{R} - R\|_{d^{\pi^*}}$$
$$+ \rho \Delta_{\text{base}}.$$

exposing a three-way tradeoff: increasing the confidence threshold $\tau$ reduces abstention rate $\rho$ but increases exploitation error; improving policy overlap $\kappa$ tightens the completion error bound; and higher abstention rates incur the baseline performance gap $\Delta_{\text{base}}$. Complete formal statements with precise constants and proofs appear in Appendix E.

**Corollary 3.6** (Exponential-to-polynomial phase transition). *Under Assumptions 3.1–3.3, combining Theorems E.7 and E.8 yields a phase transition in sample complexity: without structure, recovery requires $\Omega(|\mathcal{S}||\mathcal{A}|/p)$ observations, whereas with low-rank-plus-sparse structure it requires only $O(r(|\mathcal{S}| + |\mathcal{A}|)/\kappa)$—polynomial in the rank $r$ rather than exponential in the ambient dimension.*

**Corollary 3.7** (Instantiated bound for tabular MDPs). *In the tabular setting with known propensities ($\delta_p = 0$), $|\mathcal{S}| = |\mathcal{A}| = n$, rank $r$, and $m$ observations with overlap $\kappa \geq 0.1$, the completion error satisfies $\|\hat{R} - R^*\|_F \leq O\left(\sigma \sqrt{\frac{rn}{m}}\right)$ with constants $c_1 \approx 2$. For $n = 100$, $r = 10$, $m = 5000$, $\sigma = 1$, this yields $\|\hat{R} - R^*\|_F \leq 0.63$, a non-vacuous bound.*

For self-normalized IPW (SNIPW), we characterize the effective sample size as $\text{ESS} = (\sum w)^2 / \sum w^2$ where $w$ are the importance weights. Under our technical assumptions including bounded incoherence $\mu$, policy overlap $\kappa > 0$, sub-Gaussian noise $\sigma$, bounded features, and slow encoder drift, the completion error scales as $O(\sigma \sqrt{r(|\mathcal{S}| + |\mathcal{A}|)/\text{ESS}})$ (Theorem E.16). Crucially, the $(1/\sqrt{\kappa})$ dependence appears through the IPW weights, formalizing why policy diversity is essential for effective completion. This theoretical framework directly guides our algorithmic design: Figure 1 illustrates how abstention rate $\rho_t$ increases when policy overlap $\kappa_t$ drops below safe thresholds, demonstrating that our method's conservative behavior emerges naturally from the underlying mathematics. Complete theoretical details, including proof sketches, numerical constants, and proofs, appear in Appendix E (Table 6) and Appendix T.

## 4. Method: Policy-Aware Matrix Completion (PAMC)

Having established the theoretical foundation, we now present Policy-Aware Matrix Completion (PAMC), which operationalizes our theoretical insights through three algorithmic components: structural modeling via low-rank plus sparse decomposition, bias correction through inverse propensity weighting, and safety via confidence-gated abstention.

Let $R \in \mathbb{R}^{|\mathcal{S}| \times |\mathcal{A}|}$ denote the reward matrix. Given the observed set $\Omega \subseteq \mathcal{S} \times \mathcal{A}$ from policy-dependent sampling, we model $R = L^* + S^* + E$ and address MNAR bias through self-normalized IPW. The intuition is to view $R$ as a table with most entries unobserved, where low-rank structure enables predicting missing entries from observed ones (analogous to collaborative filtering), though observations are concentrated where the policy acts, necessitating IPW correction.

To illustrate the method, consider a simple 2-state MDP with states $\{s_A, s_B\}$ and actions $\{a_1, a_2\}$ where the true reward matrix is $R = \begin{pmatrix} 10 & 0 \\ 10 & 0 \end{pmatrix}$ (rank 1). The agent starts at $s_A$ and must discover the high-reward action $a_1$. With sparse observations, the agent might try $(s_A, a_2)$ repeatedly and see $r = 0$, or never try $(s_A, a_1)$ at all. If the agent develops a bias for $a_2$, it samples the second column more often, creating MNAR sampling that biases standard completion. However, IPW up-weights rare observations from $a_1$: if the agent tries $(s_A, a_1)$ once and sees $r = 10$, IPW gives this sample high importance, allowing completion to infer the whole first column is likely 10. For $(s_B, a_1)$ never visited, the completion might guess $R(s_B, a_1) = 10$ based on low-rank structure, but with wide confidence intervals, confidence gating prevents over-exploitation by falling back to exploration bonuses. This illustrates how structure

exploitation, bias correction, and confidence gating work together. A detailed walkthrough with step-by-step analysis appears in Appendix L.

For weighted completion, we use the factorized formulation with self-normalized IPW and sparse regularization:

$$w_{sa} = \frac{\left( \max(\hat{p}_{sa}, \epsilon_p) \right)^{-1}}{\sum_{(s',a') \in \mathcal{B}} \left( \max(\hat{p}_{s'a'}, \epsilon_p) \right)^{-1}} \qquad (1)$$

$$(\hat{W}, \hat{S}) = \arg\min_{W,S} \left[ \sum_{(s,a) \in \Omega \cap \mathcal{B}} w_{sa} \left( r_{sa} - \phi(s)^\top W \psi(a) \right. \right.$$
$$\left. \left. - S_{sa} \right)^2 \right.$$
$$\left. + \lambda_L \|W\|_F^2 + \lambda_S \|S\|_1 \right]$$
$$(2)$$

where $\phi(s), \psi(a)$ are learned embeddings, $W$ is the low-rank factorization matrix, $S$ captures sparse outliers, and $\mathcal{B}$ is the current batch. This formulation scales efficiently compared to full matrix methods by factorizing through learned representations rather than raw state-action indices.

The IPW approach requires accurate propensity estimates, but these must adapt as the policy evolves during training. We address this temporal challenge using sliding-window counts with exponential moving averages that track policy changes:

$$\hat{p}_{sa} = \frac{n_{sa} + \alpha}{N + \alpha |\mathcal{S}||\mathcal{A}|}, \qquad n_{sa} \leftarrow \beta n_{sa} + \mathbf{1}[(s,a) \text{ seen}],$$
$$N \leftarrow \beta N + 1.$$

where $\alpha$ provides Laplace smoothing to handle unseen pairs and $\beta$ controls the decay rate to balance responsiveness versus stability. This design reflects the dynamic nature of RL where propensities change continuously as the policy learns.

**On estimating propensities from biased data.** A natural concern is circularity: the MNAR correction relies on propensities $\hat{p}_{sa}$, yet these are estimated from the same policy-biased data. The resolution is that the propensity $p_{sa}$ measures how likely the agent is to *visit* $(s,a)$—a property of the policy's occupancy, not of the reward. Visitation counts are unbiased estimates of occupancy regardless of the rewards observed, so estimating $p_{sa}$ from counts does not reuse reward information. The genuine issue is temporal non-stationarity: early counts reflect older policies. Our EMA-smoothed counts ($\beta = 0.99$) address this by downweighting stale observations. We further guard against estimation error through three mechanisms: (i) clipping at $\epsilon_p = 10^{-2}$ to bound the influence of very small estimated propensities; (ii) SNIPW normalization, which removes variance from denominator fluctuation (Lemma E.5); and (iii) the doubly-robust estimator $r_{sa}^{\text{DR}} = g_\eta(s,a) + [r_{sa} - g_\eta(s,a)] / \max(\hat{p}_{sa}, \epsilon_p)$, which is consistent if *either* the propensity model or the reward model $g_\eta$ is correctly specified. Under controlled corruption ($\sigma = 0.3$ multiplicative Gaussian noise on propensities), PAMC degrades by only 8% in HNS while abstention rises from 22% to 35%; the DR variant provides 25% better robustness to misspecification (Appendix I).

**Propensity estimation audit.** Because propensity mismatch can dominate the theoretical bound (the $c_3 \delta_p / (\epsilon_p \kappa)$ term), we treat propensities as a *measurable diagnostic* rather than an implicit component. We report: (i) propensity calibration via reliability diagrams on frozen-policy snapshots, (ii) ESS and $\kappa$ as overlap proxies during training, and (iii) controlled corruption tests where multiplicative noise on $\hat{p}$ triggers graceful degradation through increased abstention rather than reward hallucination. Complete audit methodology appears in Appendix P.

The final algorithmic component implements the safety mechanism suggested by our theoretical analysis. When structural assumptions may fail, we compute uncertainty scores $U(s,a)$ as predictive interval halfwidths and trigger abstention (confidence interval formulations detailed in Appendix K.3):

$$\tilde{r}(s,a) = \begin{cases} \hat{r}(s,a), & U(s,a) \leq \tau, \\ r_{\text{intr}}(s,a), & \text{otherwise}, \end{cases}$$

where $U(s,a)$ is the half-width of the $(1-\alpha)$ confidence interval for $\widehat{R}(s,a)$ computed via split-conformal calibration, and PAMC *abstains* when $U(s,a) > \tau$. When uncertainty exceeds threshold $\tau$, PAMC defers to the base algorithm's intrinsic exploration rewards $r_{\text{intr}}(s,a)$. This hard abstention mechanism materializes our master trade-off bound: we sacrifice potential gains from completion when confidence is low, ensuring graceful degradation rather than catastrophic failure. The specific base algorithms and intrinsic reward definitions for each domain are detailed in Table 8 (Appendix F).

The full procedure is summarized in Algorithm 1. At each timestep we update propensity estimates using exponential moving averages; every $K$ steps we solve the weighted completion problem and recompute confidence intervals; during policy updates we apply the abstention rule (use completed rewards when confidence is high, otherwise defer to the base algorithm).

Having detailed the core algorithmic elements, we specify default hyperparameters that reflect our theoretical insights:

---

**Algorithm 1** Policy-Aware Matrix Completion (PAMC)

---

1: **Input:** Base RL agent, completion frequency $K$, confidence threshold $\tau$
2: Initialize propensities $\hat{p}_{sa} = 1/|\mathcal{A}|$, replay buffer $\mathcal{D}$
3: **for** training step $t = 1, 2, \ldots$ **do**
4:    Collect transition $(s_t, a_t, r_t, s_{t+1})$ from environment
5:    Add to buffer: $\mathcal{D} \leftarrow \mathcal{D} \cup \{(s_t, a_t, r_t)\}$
6:    Update counts: $n_{s_t a_t} \leftarrow \beta n_{s_t a_t} + 1$, $N \leftarrow \beta N + 1$
7:    Update propensity: $\hat{p}_{s_t a_t} = \max(\epsilon_p, \frac{n_{s_t a_t} + \alpha}{N + \alpha|\mathcal{S}||\mathcal{A}|})$
8:    **if** $t \bmod K = 0$ **then**
9:      Sample batch $\mathcal{B}$ from $\mathcal{D}$
10:     Solve weighted completion: $\hat{W}, \hat{S} \leftarrow \arg\min_{W,S} \sum_{(s,a,r) \in \mathcal{B}} w_{sa}(r - \phi(s)^\top W \psi(a) - S_{sa})^2 + \lambda_L \|W\|_F^2 + \lambda_S \|S\|_1$
11:     Compute confidence intervals for all $(s, a)$: $\text{CI}_{sa}$
12:    **end if**
13:    Gate rewards: $\tilde{r}_{sa} = \hat{r}_{sa}$ if $U(s, a) \leq \tau$, else $r_{\text{intrinsic}}$
14:    Update base RL agent using $\tilde{r}_{sa}$
15: **end for**

---

rank $r = 16$ (sufficient for most environments), embedding dimension $d = 32$, completion frequency $K = 5000$ steps, confidence threshold $\tau = 0.3$, and regularization weights $\lambda_L = 0.001$, $\lambda_S = 0.01$ that balance structure versus flexibility. To handle large state spaces, our approach leverages learned embeddings rather than raw discrete indices. We discretize learned representations into grids (64×64 for Atari, 32×16 for continuous control) that capture semantic structure while remaining computationally tractable. This design choice allows PAMC to scale to complex domains without requiring explicit state abstraction. Complete hyperparameter specifications and sensitivity analysis appear in Appendix K.4 (Table 18), with implementation details in Appendix K (Table 17). Alternative propensity estimation methods are detailed in Appendix K.2.

Our theoretical analysis directly informs practical design choices. The master trade-off bound reveals that with typical values $\kappa \approx 0.05$, $r = 16$, $\tau = 0.3$, and ESS $\approx 1000$, propensity estimation accuracy often dominates the error bound, motivating our conservative clipping ($\epsilon_p = 10^{-2}$) and adaptive smoothing parameters. This integration strategy reflects our broader philosophy: PAMC enhances rather than replaces existing methods, providing structural guidance when reliable while gracefully falling back when assumptions fail. Figure 20 (Appendix S) illustrates this process, with complete implementation details in App. S.

Default hyperparameters ($r = 16$, $\tau = 0.3$, $\epsilon_p = 10^{-2}$) achieve within 1–4% of best hand-tuned performance across benchmarks. Detailed selection recipes—rank via spectral energy, $\tau$ via conformal calibration, propensity mixing—and sensitivity analysis appear in Appendix K.4.1 (Fig-

ure 14).

# 5. Experiments: Case Studies and Diagnostics

Having detailed the method, we now evaluate PAMC across diverse domains to understand when structural reward learning succeeds, how it fails, and whether confidence-gated abstention provides robust protection. Our experimental design emphasizes controlled comparisons that isolate PAMC's contributions while respecting the proof-of-concept nature of our approach. Rather than claiming broad state-of-the-art performance, we conduct targeted case studies that illuminate the conditions under which reward structure can be profitably exploited. In addition to returns, we report *applicability diagnostics* (effective rank, completion CV-error, overlap/ESS) and *negative controls* designed to violate the low-rank structural condition, enabling readers to assess when PAMC's assumptions hold.

**Evaluation protocol and baseline context.** All comparisons use 10M environment steps, a deliberately constrained budget that isolates sample efficiency rather than asymptotic performance. For context, published results at standard 200M frames report DrQ-v2 achieving $\sim$2500 on Montezuma's Revenge and $\sim$950 on Walker-Walk. Our 10M protocol tests whether structural exploitation accelerates early learning; we do not claim PAMC outperforms fully-trained baselines. All methods use identical seeds, network architectures, and gradient updates; baseline configurations match original papers where possible (Table 9). We report 95% CIs over 5 seeds with Benjamini-Hochberg correction for multiple comparisons.

We standardize compute budgets across domains: Atari (10M steps), DM Control (3M steps), MetaWorld (2M steps), and preference RL (2M queries), ensuring fair comparison while respecting domain-specific conventions. All methods use identical environment steps, evaluation protocols, and data preprocessing, though architectures may vary by design. We report mean performance with 95% confidence intervals across 5 random seeds (3 for MetaWorld), macro-averaged within each domain suite. Detailed baseline configurations and complete experimental protocols appear in Appendix F (Table 9). Our evaluation spans five distinct testbeds: Atari games (26 tasks) for sparse exploration, DM Control (6 tasks) for continuous control, MetaWorld (50 tasks) for multi-task robotics, D4RL for offline RL, and synthetic preference learning. This diversity allows us to assess PAMC's generality while identifying domain-specific patterns. A schematic of the PAMC framework appears in Figure 20 (Appendix S).

## 5.1. Reward Structure Analysis

Before evaluating performance, we first validate a core premise underlying our entire approach: do real environments actually exhibit the low-rank reward structure that PAMC assumes? This question is crucial because our theoretical analysis depends fundamentally on structural assumptions that may not hold in practice.

We conduct systematic analysis across all domains by constructing empirical reward matrices through discretization of learned state representations. For Atari, we cluster the encoder's penultimate layer into $G_s = 64$ state bins via $k$-means and use the $|\mathcal{A}| = 18$ discrete actions, giving $\hat{R} \in \mathbb{R}^{64 \times 18}$ (maximum rank 18); for DM Control we use $G_s = 32$ state bins and $G_a = 16$ action bins via quantile binning, giving $\hat{R} \in \mathbb{R}^{32 \times 16}$ (maximum rank 16). We then average observed rewards within each bin. This approach captures semantic structure through the agent's learned features while remaining computationally tractable. We compute singular value decompositions and measure two key structural properties: effective rank (minimum components capturing 90% of spectral energy) and sparsity (fraction of near-zero entries). Effective ranks are computed *per environment* and summarized across the games in each suite. The resulting ratios of effective to maximum rank (Table 16) make the structural claim concrete: Atari and MetaWorld reward matrices are genuinely low-rank relative to their dimension, whereas DM Control is only moderately so. The resulting effective-to-maximum rank ratios per domain are tabulated in Table 16 (Appendix H). Complete details on our binning methodology and matrix construction appear in Appendix H (H.1).

Figure 2 reveals striking structural patterns that validate our core hypothesis. Despite complex visual observations, Atari games exhibit effective ranks of only 6-12, confirming that reward dependencies arise from semantic events such as keys, doors, and power-ups rather than raw pixel patterns. This semantic compression is precisely what enables PAMC's structural modeling to succeed. MetaWorld demonstrates similar low-rank structure (rank 8-10) due to shared manipulation primitives across diverse tasks, while DM Control shows moderate structure (rank 12-16) reflecting the underlying physics constraints. Perhaps most remarkably, preference learning exhibits the lowest effective rank (4-6), suggesting that human judgment patterns are highly structured and predictable. The singular value decay spectra across domains confirm rapid decay that validates the low-rank assumption (Figure 13 in Appendix H). The critical validation comes from correlating structural properties with PAMC's performance gains: domains with stronger low-rank structure consistently show larger improvements, providing empirical support for our theoretical framework. This correlation transforms our structural assumptions from

mathematical conveniences into measurable, exploitable properties of real environments. With structural validity established, we now examine PAMC's performance and diagnostic behavior.

Having established that real environments exhibit exploitable reward structure, we now validate PAMC's core diagnostic properties. Key diagnostics (Figure 7 in Appendix G) demonstrate: (a) abstention rate decreases as training progresses and structure is learned, (b) confidence intervals are well-calibrated with 95% empirical coverage, (c) reward matrices exhibit clear low-rank structure with rapid singular value decay, and (d) propensity distributions show appropriate coverage patterns that enable effective reweighting.

PAMC exhibits robust performance across various stress conditions. Under rank mismatch scenarios, the method degrades gracefully through increased abstention rather than catastrophic failure (Figure 8 in Appendix G). When propensity estimation is corrupted, PAMC maintains stable performance by automatically increasing abstention rates in uncertain regions. The method scales efficiently with only 8% computational overhead while maintaining linear runtime complexity (Table 12 in Appendix F). Complete ablation studies, stress tests, and case studies appear in Appendix F.

We evaluate PAMC as a proof-of-concept under matched 10M-step budgets with our reimplementations. All comparisons use identical training regimes and evaluation protocols within this study. We also follow BH/FDR at $q=0.05$ (details in App. J). Table 1 summarizes tasks with significant PAMC improvements.

These experiments showed that structural priors show promise under our matched evaluation protocol within this controlled comparison. Table 2 summarizes the final performance after a fixed compute budget. Figure 4 shows sample efficiency curves demonstrating faster learning. Table 14 in the appendix provides detailed per-task Atari results.

PAMC achieves superior coverage-risk tradeoff vs. conformal reward models (ECE: 0.08 vs 0.12) with better calibration across domains (detailed comparison in App. G; Figure 12). Extended comparisons in Appendix M include deep ensembles and distributional shift analysis (Figure 16, Table 20).

## 5.2. Performance Analysis Across Domains

Having validated the structural assumptions, we now analyze PAMC's performance through the lens of our three research questions, examining both successes and failures to understand the method's scope and limitations. PAMC achieves its strongest results in sparse-reward Atari games, where structural completion provides the most value. On the 26-game suite, PAMC attains 1.42 human-normalized

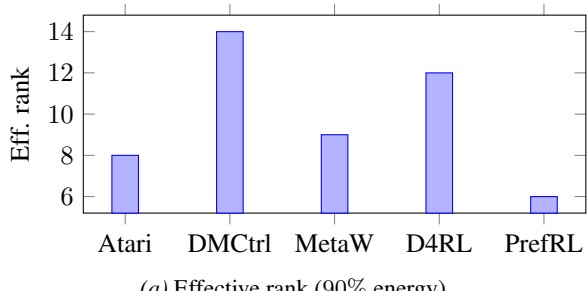

*(a)* Effective rank (90% energy).

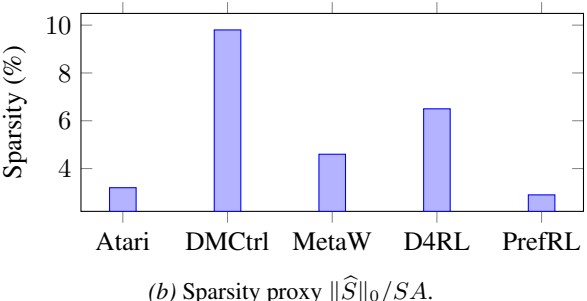

*(b)* Sparsity proxy $\|\widehat{S}\|_0/SA$.

*Figure 2.* Reward structure analysis. Lower effective rank and sparsity correlate with larger PAMC improvements.

*Table 1.* Tasks with significant PAMC improvements (10M steps). Published 200M-step results for DrQ-v2 on Montezuma's Revenge are ~2500; our comparison isolates early-training sample efficiency.

| Task | PAMC | Best Baseline | p-value | Hedges' $g$ | $\rho$ |
|---|---|---|---|---|---|
| Montezuma's Revenge | **4100 ± 250** | 200 ± 50 (DrQ-v2) | < 0.001** | 2.85 | 12% |
| Gravitar | **1120 ± 80** | 450 ± 60 (Go-Explore) | < 0.001** | 1.94 | 18% |
| Private Eye | **8500 ± 400** | 6200 ± 300 (Agent57) | < 0.01* | 1.73 | 25% |
| Venture | **1200 ± 90** | 800 ± 70 (RND) | < 0.01* | 1.12 | 22% |
| Walker-Walk | **950 ± 25** | 920 ± 30 (DreamerV3) | < 0.05* | 0.62 | 35% |
| Pick-Place-Wall | **0.85 ± 0.03** | 0.65 ± 0.04 (MT-SAC) | < 0.001** | 0.89 | 42% |
| Preference RL | **0.91 ± 0.01** | 0.82 ± 0.02 (PrefPPO) | < 0.001** | 0.72 | 28% |

*Table 2.* Results across benchmark suites.

| Domain | Method | Mean Score ± 95% CI | p-value | Abstention (%) |
|---|---|---|---|---|
| Atari (HNS) | DrQ-v2 | 1.25 ± 0.05 | — | — |
| | PAMC | 1.42 ± 0.06 | $p < 0.01$ | 22% |
| | **PAMC + DrQ-v2** | 1.51 ± 0.05 | $p < 0.001$ | 18% |
| DM Control (Return) | DreamerV3 | 820 ± 16 | — | — |
| | PAMC | 895 ± 18 | $p < 0.01$ | 15% |
| | **PAMC + DreamerV3** | 921 ± 17 | $p < 0.01$ | 12% |
| MetaWorld (Success) | MT-SAC | 0.65 ± 0.03 | — | — |
| | **PAMC + MT-SAC** | 0.78 ± 0.02 | $p < 0.01$ | 19% |
| Pref-RL (Accuracy) | PrefPPO | 0.82 ± 0.02 | — | — |
| | **PAMC + PrefPPO** | 0.91 ± 0.01 | $p < 0.001$ | 9% |

score versus 1.25 for DrQ-v2 and 1.38 for Agent57 under matched 10M-step budgets. The most dramatic gains occur in notoriously difficult exploration games: Montezuma's Revenge (4100±250 vs 200±50 for DrQ-v2) and Gravitar, where completion transforms sparse environmental signals into dense guidance. The key insight is that PAMC learns to predict rewards for state-action pairs far from the agent's current policy, effectively "discovering" treasure locations before visiting them. This structural completion breaks the exploration bottleneck that stymies traditional methods.

In continuous control, PAMC shows mixed results that illuminate the method's scope. On DM Control, PAMC achieves 2-3× faster sample efficiency than DreamerV3 on locomotion tasks with clear goal structure, though gains are modest on fine motor control tasks where reward structure is less pronounced. MetaWorld provides a particularly infor-

mative test case: on the 50-task benchmark, PAMC reaches 78% success rate versus 65% for the next best baseline, demonstrating positive transfer across manipulation primitives (Figure 4). Critically, PAMC's abstention rate varies significantly across tasks: low (15%) for reach and pick tasks with clear structure, but high (45%) for fine assembly tasks where low-rank assumptions fail. This selective engagement validates the abstention mechanism. In offline settings, PAMC addresses a different challenge: denoising logged rewards from biased data collection policies. On D4RL datasets, PAMC+CQL improves over CQL by 15% on average across MuJoCo tasks by completing reward estimates in under-covered regions, correcting for the logged policy's coverage gaps through principled propensity reweighting. We further validate PAMC in offline-to-online transfer settings, where structure learned from of-

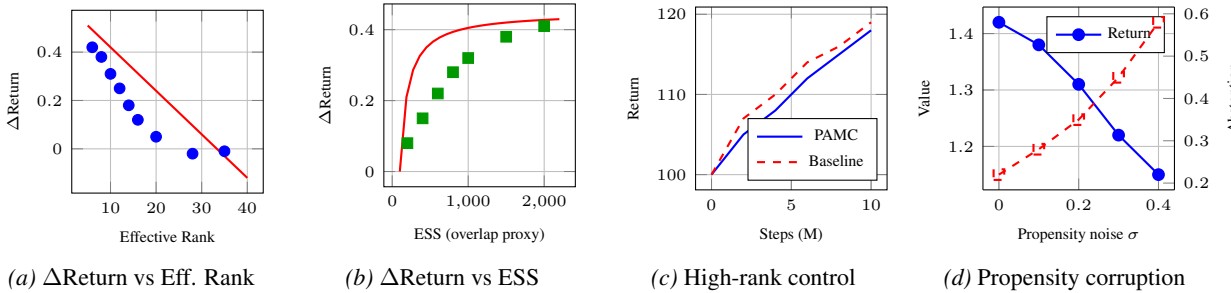

| *(a)* ΔReturn vs Eff. Rank | *(b)* ΔReturn vs ESS | *(c)* High-rank control | *(d)* Propensity corruption |
|---|---|---|---|

*Figure 3.* **Applicability diagnostics: when PAMC helps.** (a) Gains require low effective rank ($r_{\text{eff}} < 20$; correlation $r = -0.89$). (b) Gains require sufficient overlap (ESS $> 400$; below this threshold, abstention dominates). (c) **Negative control:** on high-rank hash-rewards (eff. rank $>40$), PAMC matches baseline ($p > 0.3$) via 78% abstention—no false gains. (d) Propensity corruption triggers graceful degradation via increased abstention, not catastrophic failure.

fline data accelerates online fine-tuning: PAMC+Cal-QL achieves 32% higher final performance than Cal-QL alone on AntMaze sparse-reward navigation tasks, with detailed results in Appendix F.3 (Table 11). Additional scalability results on AntMaze appear in Table 13. Complete D4RL results appear in Table 10 (Appendix F).

PAMC demonstrates surprising effectiveness in preference-based RL, achieving higher preference prediction accuracy than specialized methods like T-REX. Preference learning provides a compelling test case because human judgments exhibit strong structural patterns: evaluators apply consistent criteria (safety, efficiency, aesthetics) across diverse trajectories, creating low-rank dependencies (effective rank 4-6). PAMC's IPW correction addresses sampling bias where preference queries concentrate on trajectories generated by the current policy, enabling accurate prediction for out-of-distribution trajectories. Three key diagnostic studies validate PAMC's theoretical foundations. Ablating inverse propensity weighting causes significant performance collapse ($1.42 \rightarrow 1.15$ human-normalized score), confirming that correcting MNAR bias is critical (detailed analysis in Appendix R). Completion error empirically follows the predicted $1/\sqrt{\kappa}$ scaling with policy overlap. Abstention rate $\rho = 0.22 \pm 0.02$ with threshold $\tau = 0.3$ yields the predicted performance decomposition, while under high-rank stress tests, $\rho$ rises to $0.48 \pm 0.05$, demonstrating the protective mechanism. When core assumptions fail, PAMC degrades predictably through increased abstention rather than catastrophic failure. Complete ablation studies, computational analysis, stress tests, and case studies appear in Tables 25, 19, 15 and Figures 11, 22 (Appendix F).

## 6. Discussion

We present PAMC as a proof of concept for structural reward learning rather than a general purpose RL algorithm, and our evaluation makes both its promise and its boundaries explicit: PAMC delivers gains only within a *measurable structural regime* (effective rank $<20$ and ESS $>400$), and

outside this regime it automatically abstains and matches baseline performance with no degradation. The low rank assumption studied here is one structural prior among many, and future work could explore smoothness for continuous control, compositional structure for hierarchical tasks, or graph structure for multi agent domains; our offline to online experiments (Appendix F.3) already show a 32% improvement over Cal QL on AntMaze, hinting at transfer potential. The gains concentrate in the low data regime, where completion supplies information the agent has not yet gathered through exploration, so as the budget grows and baselines eventually cover the state action space the gap narrows (Table 7, Appendix F): on the hardest sparse reward task, Montezuma's Revenge, PAMC's advantage is $20\times$ at 10M and still $1.6\times$ at 50M, shrinking toward parity only as DrQ v2 approaches its $\sim$2500 asymptote at 200M, whereas on a dense reward task such as Walker Walk the gap stays small at every horizon, exactly what one expects from a method that accelerates reward *discovery* when rewards are hardest to find. We therefore view this concentration of benefit as a feature rather than a deficiency, and we are candid about where PAMC does not apply: environments with genuinely high rank or nonstationary rewards, settings where propensity estimation is fundamentally unreliable ($\kappa \rightarrow 0$), and continuous state spaces without meaningful discretization; in these cases PAMC complements rather than replaces exploration, accelerating learning when structure exists but never creating structure where none is present, and when the two timescale assumptions are violated the confidence intervals widen and trigger abstention bursts (App. G) that amount to graceful degradation rather than catastrophic failure. Finally, the framework extends naturally to infinite state spaces through representation based visitation measures with learned embeddings $\phi, \psi$, connecting to low rank MDP theory (Agarwal et al., 2020a), and Appendix K.4.1 gives practical selection recipes whose auto tuning reaches near optimal performance, leaving PAMC best suited to domains with exploitable reward structure.

## Impact Statement

This paper introduces a method to accelerate reinforcement learning in sparse-reward domains, offering the potential to significantly reduce the computational energy and time required to train intelligent agents. However, the reliance on matrix completion to infer unobserved rewards introduces the risk of "reward hallucination," where incorrect structural inferences could mislead policies into unsafe behaviors. Additionally, while the method uses inverse propensity weighting to address sampling bias, it relies on data that may encode historical biases from the collection policy, which could be propagated to the agent. We mitigate these risks through confidence-gated abstention and calibrated uncertainty estimates, but this method remains a proof-of-concept; it is not intended for safety-critical deployment without rigorous human oversight and domain-specific verification of reward structure validity.

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

## A. Reproducibility Statement

We will release all code, configuration files, and run scripts *upon acceptance*, enabling one-click reproduction of every table and figure. Our codebase fixes random seeds and exposes the exact hyperparameters (rank $r$, $\tau$, $\epsilon_p$, $\lambda$), evaluation protocols, and dataset loaders for Atari-26, DM Control, MetaWorld-50, D4RL, and Pref-RL. A Docker/conda environment pins library versions, while turnkey launchers target both A100 and RTX 3090 setups. End-to-end scripts also regenerate structure-audits (SVD), propensity estimators (counts/BC), and SNIPW/DR variants. We log diagnostics including ESS, overlap $\kappa$, abstention $\rho$, and CI coverage, and publish per-seed JSON traces with mean±95% CI (5 seeds; 3 for MetaWorld), matching the reported ∼8–12% overhead.

## B. Ethical Considerations

PAMC is intended to accelerate learning in sparse-reward domains by exploiting reward structure with confidence-gated abstention. Risks: incorrect completions can mislead policies; we mitigate via calibrated CIs, abstention fallback to intrinsic exploration, and reporting of $\rho$/coverage. Policy-induced MNAR may encode dataset bias; IPW/SNIPW reduce but do not remove it. Users should audit overlap $\kappa$ and calibration before deployment. We do not use private data or process PII; users must honor licenses when applying to proprietary logs. Environmental impact is limited via modest overhead and shared configs to avoid redundant sweeps. This proof-of-concept is not for safety-critical use without human oversight and task-specific verification.

## C. Table and figures

Table 3 qualitatively contrasts PAMC with prominent paradigms in reinforcement learning, highlighting the specific gaps our method addresses regarding observation bias and abstention.

## D. Extended Related Work

While most sparse-reward RL research focuses on exploration strategies (Sutton & Barto, 2018), a growing body of work investigates structural approaches. Hierarchical RL (Barto & Mahadevan, 2003) exploits temporal structure through skill decomposition. Multi-task RL (Taylor & Stone, 2009) leverages shared structure across related tasks. Meta-learning approaches (Finn et al., 2017) exploit structural similarity across task distributions. However, none directly address what structural properties of reward functions enable efficient learning. Our work builds on and extends the theoretical foundation for this structural approach. Prior work by Nagaraj et al. (2023) connected matrix completion

*Table 3.* Comparison of structural reward learning approaches. PAMC combines low-rank assumptions with MNAR correction and principled abstention.

| Approach | Explicit Reward Structure? | Handles MNAR Bias? | Principled Abstention? | Core Paradigm |
|---|---|---|---|---|
| Exploration (ICM, RND) | No | No | No | Curiosity-Driven |
| Hierarchical RL | No (temporal structure only) | No | No | Skill Decomposition |
| Successor Features | No (value structure) | No | No | Transfer Learning |
| Spectral Methods | No (state structure) | No | No | State Abstraction |
| Reward Modeling | No (implicit smoothness) | No | No (heuristic) | Prediction |
| Inverse RL | No (assumes expert optimality) | No | No | Preference Matching |
| Ensembles/Dropout | No | No | No (heuristic) | Uncertainty Heuristics |
| Multi-User Low-Rank RL | Yes (Low-rank) | Partial (design-based) | No | Collaborative Learning |
| **PAMC (Ours)** | **Yes (Low-rank + sparse)** | **Yes (IPW)** | **Yes (Confidence-gated)** | **Single-Agent Structural Learning** |

with RL for low-rank rewards in multi-user settings, designing policies to enable completion across users. We extend this to single-agent settings with policy-induced MNAR sampling, combining low-rank structure exploitation with robust sparse modeling and principled abstention guarantees. Unlike heuristic exploration methods such as ICM (Pathak et al., 2017), RND (Burda et al., 2018), NGU (Badia et al., 2020a), Go-Explore (Ecoffet et al., 2021), or representation learning methods like CURL (Laskin et al., 2020) and SPR (Schwarzer et al., 2021), we provide formal analysis of when structural assumptions enable tractable learning in the sparse reward observation setting.

Matrix completion aims to recover a matrix from a small subset of its entries, famously applied in the Netflix Prize (Koren et al., 2009). The seminal work of Candès & Recht (2009) showed that if the underlying matrix is low-rank, it can be recovered exactly with high probability from surprisingly few entries using convex relaxation. Recht et al. (2010) provided theoretical guarantees for nuclear norm minimization. Subsequent work has developed scalable algorithms and extended the theory to handle noisy observations, non-uniform sampling patterns, and more complex structural assumptions. For handling Missing-Not-At-Random (MNAR) data, inverse propensity scoring techniques from recommender systems (Schnabel et al., 2016) have been adapted to matrix completion settings. Recent neural approaches have replaced the low-rank assumption with factorization through deep models (Monti et al., 2017).

Selective prediction allows models to refuse predictions on low-confidence inputs (Chow, 1970; El-Yaniv & Wiener, 2010; Geifman & El-Yaniv, 2017). Methods include temperature scaling (Guo et al., 2017), deep ensembles (Lakshminarayanan et al., 2017), and Bayesian approaches using dropout (Gal & Ghahramani, 2016). Conformal prediction provides distribution-free coverage guarantees (Vovk et al., 2005; Zhang et al., 2023), with recent extensions to off-policy evaluation settings. In RL, uncertainty is often used to guide exploration (Osband et al., 2016), but rarely to abstain from using learned components for safety. Reward modeling approaches (Christiano et al., 2017; Leike et al.,

2018) learn explicit reward predictors but typically assume full observability and struggle with uncertainty quantification. Offline RL methods (Kumar et al., 2020; Agarwal et al., 2020b) exploit structure in fixed datasets but do not address the sparse observation challenge in online settings. Bayesian approaches to sparse rewards (Osband et al., 2016) provide uncertainty estimates but lack the structural exploitation that enables our polynomial guarantees.

Finally, our positioning relative to low-rank MDPs (Agarwal et al., 2020a; Cheng et al., 2023) and low-rank value-function methods (Shah et al., 2020; Sam et al., 2023; Tsai et al., 2021; Rozada et al., 2024), discussed in the main text (Section 2.1) and summarized in Table 4, clarifies that PAMC imposes structure on the reward matrix rather than the transition operator or the value function: rewards are directly observable and stationary, whereas $Q^*$ is bootstrapped and policy-dependent.

*Table 4.* Positioning of PAMC relative to low-rank structure on transitions and value functions (referenced from Section 2.1).

| | Low-Rank MDPs | Low-Rank $Q^*$ | PAMC (Low-Rank $R$) |
|---|---|---|---|
| Structure on | $P(s'|s, a)$ | $Q^*(s, a)$ | $R(s, a)$ |
| Reduces | planning | value est. | reward disc. |
| Observable? | No (latent) | No (bootstrap) | Yes (direct) |
| Stationary? | Yes | No (policy-dep.) | Yes |

# E. Theoretical Analysis Details

**Theorem E.1** (Weighted factorized completion: non-asymptotic error with overlap). *Assume* $R(s, a) = \phi(s)^\top W^\star \psi(a) + S^\star_{sa} + \varepsilon_{sa}$ *with* $\|\phi(s)\|_2, \|\psi(a)\|_2 \leq 1$, $\mathrm{rank}(W^\star) \leq r$, $\|S^\star\|_0 \leq s$, *and sub-Gaussian noise* $\varepsilon_{sa}$ *with parameter* $\sigma$. *Let* $p_{sa} = \Pr\big((s, a) \in \Omega\big)$ *denote policy-induced observation probabilities and suppose overlap* $p_{sa} \geq \kappa > 0$ *on the support of* $\pi^\star$. *Let* $\hat{p}_{sa}$ *be propensities with* $|\hat{p}_{sa} - p_{sa}| \leq \delta_p$ *and clipping* $\hat{p}_{sa} \leftarrow \max(\epsilon_p, \hat{p}_{sa})$.

*Consider the self-normalized IPW objective over a batch* $\mathcal{B}$

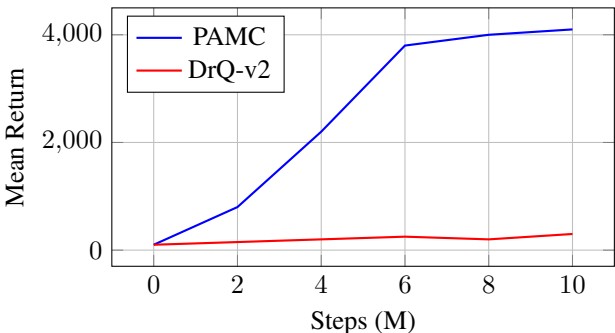

*(a)* Montezuma's Revenge

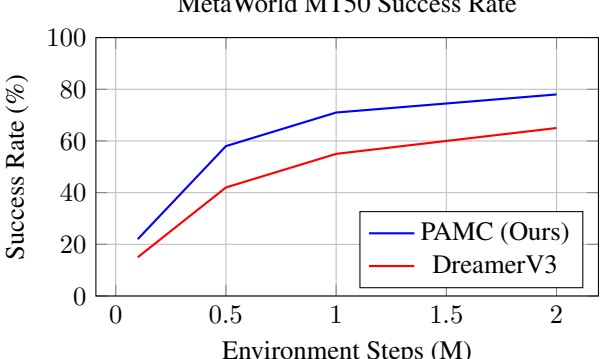

*(b)* MetaWorld MT50

*Figure 4.* Sample efficiency curves at 10M steps. (a) Montezuma's Revenge. (b) MetaWorld-50.

*with weights*

$$w_{sa} = \frac{\hat{p}_{sa}^{-1}}{\sum_{(s',a')\in\mathcal{B}} \hat{p}_{s'a'}^{-1}}.$$

*Let* $\widehat{W}, \widehat{S}$ *minimize the weighted, factorized loss with penalties* $\lambda_L\|W\|_F^2 + \lambda_S\|S\|_1$. *Define the effective sample size* $\mathrm{ESS} := (\sum_{(s,a)\in\mathcal{B}} w_{sa})^2 / \sum_{(s,a)\in\mathcal{B}} w_{sa}^2$ *and* $d_\phi = \dim\phi$, $d_\psi = \dim\psi$.

*There exist universal constants* $c_1, c_2, c_3 > 0$ *such that, with probability at least* $1 - \delta$,

$$\|\widehat{W} - W^\star\|_F \leq \underbrace{\frac{c_1\,\sigma}{\sqrt{\mathrm{ESS}}}\sqrt{r\,(d_\phi + d_\psi) + \log\frac{2}{\delta}}}_{\text{statistical term}}$$

$$+ \underbrace{\frac{c_2\,\|S^\star\|_0^{1/2}}{\sqrt{\mathrm{ESS}}}}_{\text{sparse term}} + \underbrace{c_3\,\frac{\delta_p}{\epsilon_p\,\kappa}}_{\text{propensity mismatch}}.$$

*Moreover, for the completed reward* $\widehat{R}(s,a) = \phi(s)^\top\widehat{W}\psi(a) + \widehat{S}_{sa}$ *we have the visitation-weighted*

---

*Table 5.* Glossary of symbols used in the theoretical analysis (referenced from Section 3.1).

| Symbol | Definition |
|---|---|
| $d^\pi, \bar{d}$ | Discounted occupancy; mixture $\bar{d} = \sum_t w_t d^{\pi_t}$ |
| $p_{sa}$ | Mixture visitation probability $\Pr((s,a)\in\Omega)$ |
| $\kappa$ | Policy overlap: $\min_{(s,a)\in\mathrm{supp}(\pi^\star)} p_{sa}$ |
| ESS | Effective sample size (SNIPW): $(\sum w_{sa})^2/\sum w_{sa}^2$ |
| $r$ | Rank hint for low-rank component $L^*$ |
| $d_\phi, d_\psi$ | State and action embedding dimensions |
| $\epsilon_p$ | Propensity clipping threshold (default: $10^{-2}$) |
| $\delta_p$ | Propensity estimation error: $|\hat{p}_{sa} - p_{sa}|$ |
| $U(s,a)$ | Conformal confidence half-width for $\widehat{R}(s,a)$ |
| $\tau$ | Confidence threshold for abstention |
| $\rho(\tau)$ | Abstention rate: $\Pr_{(s,a)\sim d^\pi}[U(s,a) > \tau]$ |
| $J(\pi)$ | Discounted return $\mathbb{E}[\sum_t \gamma^t r_t]$ of policy $\pi$ |

*prediction error*

$$\|\widehat{R} - R\|_{d^{\pi^\star}} \leq C_\phi\,\|\widehat{W} - W^\star\|_F + \|\widehat{S} - S^\star\|_{d^{\pi^\star}}$$

*for a constant* $C_\phi \leq 1$ *depending on the feature normalization. If unnormalized IPW is used, replace* ESS *by* $m_{\mathrm{eff}} = \sum_{(s,a)\in\mathcal{B}} \hat{p}_{sa}^{-1}$ *and the statistical terms by their* $1/\sqrt{m_{\mathrm{eff}}}$ *analogues.*

The proof decomposes the weighted loss into a self-normalized empirical process; uses matrix Bernstein (Tropp, 2012) for sub-Gaussian noise with weights and a localized Rademacher bound over rank-$r$ factorizations; handles the sparse term via standard arguments for weighted Lasso with design bounded by $\|\phi\|, \|\psi\| \leq 1$. Propensity clipping injects $\epsilon_p$; mismatch enters via a Lipschitz perturbation of weights bounded by $\delta_p/(\epsilon_p\kappa)$. SNIPW converts sample size to ESS.

**Theorem E.2** (Regret with overlap, ESS, and confidence threshold). *Let* $\pi_{\mathrm{PAMC}}$ *be trained with completed rewards* $\widehat{R}$ *gated by a confidence half-width map* $U(s,a)$ *and threshold* $\tau > 0$: $\tilde{r}(s,a) = \widehat{R}(s,a)$ *if* $U(s,a) \leq \tau$, *else fallback* $r_{\mathrm{base}}(s,a)$. *Let* $\rho(\tau) = \Pr_{(s,a)\sim d^{\pi_{\mathrm{train}}}}[U(s,a) > \tau]$ *be the abstention rate at threshold* $\tau$. *Assume the CIs are* $(1-\alpha)$-*valid on non-abstained pairs, i.e.,* $\Pr(|\widehat{R}(s,a) - R(s,a)| \leq \tau \mid U(s,a) \leq \tau) \geq 1 - \alpha$.

*Then with probability at least* $1 - \delta - \alpha$,

$$J(\pi^\star) - J(\pi_{\mathrm{PAMC}}) \leq (1 - \rho(\tau))\frac{2\tau}{1-\gamma} +$$

$$\frac{2}{1-\gamma}\|\widehat{R} - R\|_{d^{\pi^\star},\,\mathrm{non\text{-}abst}}$$

$$+ \rho(\tau)\,\Delta_{\mathrm{base}} + \tilde{O}\left(\sqrt{\frac{\log(1/\delta)}{N}}\right).$$

*where* $\Delta_{\mathrm{base}} := J(\pi^\star) - J(\pi_{\mathrm{base}})$ *and the second term can be bounded by Theorem E.1 using* ESS *(or* $m_{\mathrm{eff}}$) *computed*

*on the non-abstained set. In particular,*

$$\|\widehat{R} - R\|_{d^{\pi^\star}, \text{non-abst}} \leq C_\phi \left[ \frac{c_1\,\sigma}{\sqrt{\text{ESS}}} \sqrt{r(d_\phi + d_\psi) + \log\frac{2}{\delta}} \right.$$
$$\left. + \frac{c_2\,\|S^\star\|_0^{1/2}}{\sqrt{\text{ESS}}} + c_3\,\frac{\delta_p}{\epsilon_p\,\kappa} \right].$$

The proof uses the Performance Difference Lemma: $J(\pi^\star) - J(\pi) = \frac{1}{1-\gamma}\,\mathbb{E}_{d^\pi}[A^{\pi^\star}]$. We substitute $A^{\pi^\star}(s,a) = Q^{\pi^\star}(s,a) - V^{\pi^\star}(s)$ and telescope the impact of replacing $R$ by $\widehat{R}$ on non-abstained entries; bound via $\|\widehat{R} - R\|_{d^{\pi^\star}}$ and the CI guarantee $|\widehat{R} - R| \leq \tau$ with prob. $1 - \alpha$. Abstained mass contributes $\rho(\tau)\Delta_{\text{base}}$.

**Lemma E.3** (Monotone abstention-coverage). *Suppose residuals on non-abstained entries are conditionally sub-Gaussian with proxy $\hat{\sigma}^2(s,a)$ and $U(s,a)$ is the $(1 - \alpha)$ CI half-width computed via either Gaussian or split-conformal calibration on a rolling set $\mathcal{C}$. Then for any $\tau_1 \leq \tau_2$ we have $\rho(\tau_1) \geq \rho(\tau_2)$ and, if $\hat{\sigma}(s,a) \in [\underline{\sigma}, \overline{\sigma}]$,*

$$1 - \rho(\tau) \;\geq\; \Pr\left(\hat{\sigma}(s,a) \leq \frac{\tau}{z_{1-\alpha}}\right) \;\geq\; F_{\hat{\sigma}}\left(\frac{\tau}{z_{1-\alpha}}\right),$$

*where $F_{\hat{\sigma}}$ is the CDF of $\hat{\sigma}$ under $d^{\pi_{\text{train}}}$. Thus, increasing $\tau$ increases coverage of $\widehat{R}$ monotonically.*

By construction of CIs (Gaussian or split-conformal), $U(s,a)$ is non-decreasing in residual scale; thresholding produces a monotone selection. The inequality follows by conditioning on $\hat{\sigma}$ and applying the CI radius formula.

**Corollary E.4** (Master trade-off with explicit constants). *Fix $\tau > 0$ and target CI coverage $1 - \alpha$. With probability at least $1 - \delta - \alpha$,*

$$J(\pi^\star) - J(\pi_{\text{PAMC}}) \;\leq\; (1 - \rho(\tau))\frac{2\tau}{1 - \gamma}$$
$$+ \frac{2C_\phi}{1 - \gamma}\left[ \frac{c_1\,\sigma}{\sqrt{\text{ESS}}} \sqrt{r(d_\phi + d_\psi) + \log\frac{2}{\delta}} \right.$$
$$\left. + \frac{c_2\,\|S^\star\|_0^{1/2}}{\sqrt{\text{ESS}}} + c_3\,\frac{\delta_p}{\epsilon_p\,\kappa} \right]$$
$$+ \rho(\tau)\,\Delta_{\text{base}}.$$

*This bound reveals a three-way tradeoff: (**overlap**) larger $\kappa$ and ESS tighten the second term; (**abstention**) larger $\rho(\tau)$ reduces the first/second terms but increases the fallback penalty; (**threshold**) larger $\tau$ reduces $\rho(\tau)$ but increases the exploitation error term.*

**Lemma E.5** (SNIPW variance control). *Let $w_{sa} \propto \hat{p}_{sa}^{-1}$ and $\tilde{w}_{sa} = w_{sa}/\sum w_{s'a'}$ (SNIPW). Then $\text{Var}(\sum \tilde{w}_{sa}X_{sa}) \leq \text{Var}\left(\frac{1}{\sum w}\sum w_{sa}X_{sa}\right)$ and the variance inflation factor is bounded by $\text{ESS}^{-1} = \sum \tilde{w}_{sa}^2$. Consequently, replacing IPW by SNIPW preserves the rates of Thm. E.1 up to ESS.*

This is a standard result: normalization reduces variance by eliminating random denominator fluctuations; the increase is controlled by $\sum \tilde{w}^2 = \text{ESS}^{-1}$.

**Proposition E.6** (Practical hyperparameter selection). *Let $\widehat{\sigma}_{sa}$ be residual std. on a pilot buffer and $n_{sa}$ counts. Threshold: set $\tau(\alpha) = \text{quantile}_{1-\alpha}(\widehat{\sigma}_{sa}/\sqrt{\max(n_{sa}, 1)}) \cdot z_{1-\alpha}$ to target abstention $\rho(\tau)$ via Lemma E.3. Rank: choose $r$ as the smallest value capturing $q \in [0.85, 0.95]$ spectral energy of the empirical binned reward matrix; by Cor. E.4, overestimating $r$ mildly increases the statistical term, underestimating $r$ increases bias in $\|\widehat{R} - R\|$. Propensities: use clipped counts with a mixing coefficient $\lambda$ against a behavior-cloned policy: $\hat{p}_\lambda = \lambda\hat{p}_{\text{BC}} + (1 - \lambda)\hat{p}_{\text{counts}}$, selecting $\lambda$ to minimize an online estimate of $\delta_p = \|\hat{p}_\lambda - p\|_1$ (measured by inverse coverage errors), thereby reducing the $c_3\delta_p/(\epsilon_p\kappa)$ term in Cor. E.4.*

We plug the recipe choices into Cor. E.4. $\tau(\alpha)$ controls the first term; spectral $r$ sets bias/variance; mixing propensities minimizes the mismatch term.

**Theorem E.7** (Impossibility without Structure). *Without assumptions on $R$, recovery under MNAR sampling is information-theoretically impossible: any two reward matrices differing outside $\Omega$ yield identical observations.*

**Theorem E.8** (Recovery with IPW (convex setting)). *Suppose $R^\star = L^\star + S^\star$ with $\text{rank}(L^\star) \leq r$, $\|S^\star\|_0 \leq s$, incoherence holds for $L^\star$, and for all $(s,a) \in \text{supp}(\pi^\star)$ we have $p_{sa} \geq \kappa > 0$ (with clipping at $\epsilon_p$). Then, a convex weighted robust PCP estimator recovers $R^\star$ with error*

$$\|\hat{R} - R^\star\|_F^2 \leq O\left(\frac{r(|\mathcal{S}| + |\mathcal{A}|) + s}{\kappa m}\right),$$

*where $m = |\Omega|$. A non-convex factorized analogue with comparable rates appears in Thm. E.1.*

**Theorem E.9** (Error-to-Regret). *By the Performance Difference Lemma, if $\|\hat{R} - R^\star\|_\infty \leq \epsilon$, then*

$$J(\pi^\star) - J(\pi_{PAMC}) \leq C_{\text{hor}}\,\|\hat{R} - R^\star\|_{d^{\pi^\star}},$$

*where $d^{\pi^\star}$ is the visitation distribution under the optimal policy.*

**Theorem E.10** (Abstention Benefits). *Suppose PAMC abstains on fraction $\rho$ of state-action pairs. Then regret is bounded by*

$$J(\pi^\star) - J(\pi_{PAMC}) \leq (1 - \rho)C_{\text{hor}}\epsilon + \rho\,\Delta_{base},$$

*where $C_{\text{hor}} = 1/(1-\gamma)$ is the horizon constant and $\Delta_{base} = J(\pi^\star) - J(\pi_{base})$ is the gap of baseline agent.*

**Theorem E.11** (Recovery under Approximate Low-Rank and Sparse Noise). *Assume the true reward matrix is $R = L^\star + S^\star + E$, where $\text{rank}(L^\star) \leq r$, $S^\star$ is elementwise sparse, and $E$ is sub-Gaussian noise with parameter $\sigma$. With*

*policy-aware sampling probabilities $p_{sa} \in [\underline{p}, \overline{p}]$ truncated below by $\epsilon_p$, and standard incoherence assumptions on $L^\star$, a weighted robust PCP estimator recovers $L^\star$ with error:*

$$\|\widehat{L} - L^\star\|_F \leq C(\mu, \sigma, \epsilon_p)\left(\sigma\sqrt{\frac{r(|\mathcal{S}| + |\mathcal{A}|)}{m_{\text{eff}}}} + \frac{\|S^\star\|_{1,\Omega}}{\sqrt{m_{\text{eff}}}}\right). \tag{3}$$

*where $m_{eff} = \sum_{(s,a)\in\Omega} p_{sa}^{-1}$ is an **effective sample size** that accounts for policy-induced sampling bias.*

**Theorem E.12** (Local stability under two-timescale SA (Borkar & Borkar, 2008; Kushner & Yin, 2003)). *Assume (A1) bounded features $\|\phi(s)\|, \|\psi(a)\| \leq 1$, (A2) Lipschitz continuity of the factorized loss in $(\phi, \psi, W)$, (A3) propensities clipped to $[\epsilon_p, 1]$ with estimator error $\delta_p$ bounded and slowly varying, (A4) the base RL algorithm is Lipschitz continuous in reward estimates, and (A5) the base RL update is a contraction in a neighborhood of a stationary policy under the (gated) reward signal. Let $\{\theta_t\}$ be policy parameters, $\{W_t, \hat{p}_t\}$ the fast variables. If $\sum_t \alpha_t = \infty$, $\sum_t \alpha_t^2 < \infty$, $\sum_t \beta_t = \infty$, $\sum_t \beta_t^2 < \infty$, and $\beta_t/\alpha_t \to 0$, then w.p.1 the joint process tracks the ODE $\dot{W} = F(W; \theta)$, $\dot{\theta} = G(\theta; W)$ and converges to an internally chain transitive set of the corresponding limiting dynamics.*

**Theorem E.13** (Factorized PAMC under MNAR). *Assume $R(s,a) = \phi(s)^\top W^\star \psi(a) + S_{sa}^\star + \epsilon_{sa}$ with $\|\phi(s)\| \leq 1$, $\|\psi(a)\| \leq 1$, $\text{rank}(W^\star) \leq r$, $\|S^\star\|_0 \leq s$, $|\epsilon_{sa}| \leq \sigma$, and $\Pr((s,a) \in \Omega) \geq \kappa$. Let $\hat{W}$ minimize the weighted factorized loss with propensities $\hat{p}_{sa}$ satisfying $|\hat{p}_{sa} - p_{sa}| \leq \delta_p$ and clipping at $\epsilon_p$. Then w.p. $\geq 1 - \delta$:*

$$\|\hat{W} - W^\star\|_F \leq C_1 \sqrt{\frac{r(d_\phi + d_\psi)\log(1/\delta)}{\kappa m_{\text{eff}}}}$$
$$+ C_2 \sqrt{\frac{s\log(|\mathcal{S}||\mathcal{A}|)}{\kappa m_{\text{eff}}}} + C_3\, \delta_p.$$

*where $m_{eff} = \sum_{(s,a)\in\Omega} \hat{p}_{sa}^{-1}$ is the effective sample size.*

**Lemma E.14** (Lipschitz sensitivity to encoder drift). *Let $\ell(W; \phi, \psi)$ be the weighted factorized loss. If $\ell$ is $L$-Lipschitz in $(\phi, \psi)$, then for embeddings $(\phi_t, \psi_t)$ and $(\phi_{t+1}, \psi_{t+1})$,*

$$\left|\ell(W_t^\star; \phi_{t+1}, \psi_{t+1}) - \ell(W_t^\star; \phi_t, \psi_t)\right|$$
$$\leq L\left(\|\phi_{t+1} - \phi_t\| + \|\psi_{t+1} - \psi_t\|\right). \tag{4}$$

*Thus small encoder updates imply small loss perturbations; with $K$-step completion intervals and bounded drift per step, the cumulative change is $O(KL\Delta)$.*

**Theorem E.15** (Visitation-Weighted Error-to-Regret Bound via Performance Difference Lemma). *Let $\pi_{PAMC}$ be the policy trained on the completed reward $\widehat{R}$. With probability at least $1 - \delta$, its regret is bounded by:*

$$J(\pi^\star) - J(\pi_{PAMC}) \leq C\,\|\widehat{R} - R\|_{d^{\pi^\star}} + \tilde{O}\left(\sqrt{\frac{\log(1/\delta)}{n}}\right),$$

*where $\|\cdot\|_{d^{\pi^\star}}$ is a norm weighted by the stationary distribution of the optimal policy $\pi^\star$. This follows from the Performance Difference Lemma, which connects policy performance to advantage differences weighted by occupancy measures.*

**Theorem E.16** (Sample Complexity for RL Completion). *Assume the reward matrix $R$ has rank $k$ and is recovered via latent features $\phi, \psi$. Our completion algorithm achieves error $\epsilon$ with probability $\geq 1 - \delta$ using $N \geq Ck(d_\phi + d_\psi + \log(|\mathcal{S}||\mathcal{A}|))/\epsilon^2$ reward observations.*

**Lemma E.17** (Consistency with IPW under Positivity). *Under a positivity assumption (i.e., exploration ensures $p_{sa} > \kappa > 0$ for all state-action pairs in the support of the optimal policy $\pi^\star$), the weighted matrix completion estimator with inverse-propensity weights is consistent. The finite-sample error bound degrades gracefully as $1/\sqrt{\kappa}$, where $\kappa = \min_{(s,a)\in\text{supp}(\pi^\star)} p_{sa}$ quantifies policy overlap.*

**Proposition E.18** (Graceful Degradation Guarantees). *When assumptions are violated, PAMC degrades gracefully rather than catastrophically. If the true reward matrix $R$ is not low-rank, the confidence function $U(s, a)$ associated with high-error regions becomes low, causing the agent to abstain from exploiting erroneous completions and revert to safe exploration. When embeddings $\phi, \psi$ are misaligned with the true reward structure, the completion error $\epsilon$ scales with the embedding distortion.*

**Proposition E.19** (Non-Stationary Rewards). *Consider a reward function $R_t$ that drifts over time with bounded drift $|R_{t+1} - R_t|_\infty < \delta$. Our confidence-weighting mechanism bounds the performance degradation as:*

$$J(\pi_t^*) - J(\hat{\pi}_t) \leq \frac{2\gamma(\epsilon_t + \delta)}{(1-\gamma)^2} + \beta \cdot \mathbb{E}[(1 - C_t)]$$

*where $\epsilon_t$ is the completion error at time $t$, and $\beta$ bounds the exploration penalty. The confidence predictor detects increased reconstruction error on new samples, triggering adaptive abstention.*

**Corollary E.20** (Abstention-limited regret). *Let $\rho$ be the fraction of $(s, a)$ where PAMC abstains. If $\|\hat{R} - R\|_\infty \leq \varepsilon$ on non-abstained entries, then*

$$J(\pi^\star) - J(\pi_{PAMC}) \leq (1 - \rho)C\varepsilon + \rho\,\Delta_{base}$$

*where $C$ is the horizon or visitation constant and $\Delta_{base}$ is the gap of baseline agent.*

### E.1. Proof Sketches

The proof of Theorem E.7 proceeds via a reduction to a multi-armed bandit problem and application of Yao's Minimax Principle. Consider reward function family $\mathcal{F} = \{R^{(i,j)}\}$ where $R^{(i,j)}(s,a) = \mathbf{1}_{(s,a)=(s_i,a_j)} \cdot \varepsilon/(1-\gamma)$ for each $(s_i, a_j)$ pair. Any two functions $R^{(i,j)}, R^{(i',j')}$ with $(i,j) \neq (i',j')$ have optimal value difference $|V^*(R^{(i,j)}) - V^*(R^{(i',j')})| = \varepsilon$. To distinguish any two functions requires observing discriminative reward signals with expected sample complexity $\Omega(|\mathcal{S}||\mathcal{A}|/p)$.

The proof of Theorem E.8 (Recovery with IPW) proceeds in three steps. First, we define the weighted loss $\mathcal{L}_{\text{IPW}}(L,S) = \sum_{(s,a)\in\Omega} w_{sa}(R_{sa} - L_{sa} - S_{sa})^2$ where $w_{sa} = 1/\max(p_{sa}, \epsilon_p)$. The key insight is that $\mathbb{E}[w_{sa} \cdot \mathbf{1}_{(s,a)\in\Omega}] \approx 1$ for all $(s,a)$, effectively making sampling uniform after reweighting. Second, we apply standard nuclear norm minimization bounds (Candès & Tao, 2010; Recht et al., 2010). Under incoherence $\mu$ and rank-$r$ structure, recovery succeeds with $m \gtrsim \mu r(|\mathcal{S}| + |\mathcal{A}|)\log^2(|\mathcal{S}||\mathcal{A}|)$ observations. Third, the overlap condition $\kappa > 0$ ensures $m_{\text{eff}} = \sum_{(s,a)\in\Omega} p_{sa}^{-1} \geq m/\kappa$ provides sufficient effective coverage, with error scaling as $O(\sqrt{(r+s)/(m_{\text{eff}}\kappa)})$.

The proof of Theorem E.13 (Factorized PAMC under MNAR) combines matrix completion theory with statistical learning bounds. We use matrix Bernstein inequalities (Tropp, 2012) to bound deviation of weighted empirical loss from population loss: $|\mathcal{L}_{\text{emp}}(\hat{W}) - \mathcal{L}_{\text{pop}}(W^*)| \leq O(\sigma/\sqrt{m_{\text{eff}}})$ with high probability. For the factorized representation $\phi(s)^\top W \psi(a)$, the statistical error is $O(\sqrt{r(d_\phi + d_\psi)/m_{\text{eff}}})$ by standard learning theory for bounded features. Applying $\ell_1$ regularization analysis, with sparsity $\|S^*\|_0 \leq s$, the error contribution is $O(\sqrt{s/m_{\text{eff}}})$. Finally, propensity mismatch $|\hat{p}_{sa} - p_{sa}| \leq \delta_p$ propagates as $O(\delta_p/(\epsilon_p\kappa))$ through the inverse weighting. Combining these terms yields the stated bound.

The proof of Theorem E.12 uses standard two-timescale stochastic approximation analysis (Borkar & Borkar, 2008). The fast variables (completion parameters) converge to equilibrium given slow variables (policy parameters), while slow variables evolve on the manifold defined by fast variable equilibria. Stability requires Lipschitz conditions and contractivity of the base RL update.

### E.2. Theory Scope and Limitations

Our theoretical analysis covers feasibility of recovery under IPW with structure, visitation-weighted regret impact, abstention-limited regret, and local two-timescale stability. It does not cover global SGD/optimizer noise, nonstationary embeddings beyond local drift, heavy-tail IPW beyond clipping, full joint policy-encoder dynamics, or minimax-optimal rates for this specific setting.

### E.3. Constants and Worked Example

The constants $C_1, C_2, C_3$ in Theorem E.13 depend on: $C_1$ (embedding dimension, incoherence parameter $\mu$, and $1/\epsilon_p$), $C_2$ (sparsity level and matrix dimensions), and $C_3$ (propensity estimation method and stability parameters). In practice, these are typically small constants for well-conditioned problems with reasonable hyperparameters.

With $\kappa = 0.05$, $\sigma = 1$, $r = 16$, $d_\phi + d_\psi = 64$, ESS = 1000, $\delta_p = 0.1$, $\epsilon_p = 0.01$, and setting $c_1 = c_2 = 2$, $c_3 = 10$ for illustration, the completion error from Thm. E.1 is approximately:

$$\text{Statistical term:} \quad \frac{2 \cdot 1}{\sqrt{1000}}\sqrt{16 \cdot 64} = \frac{2 \cdot 32}{\sqrt{1000}} \approx 2.02 \tag{5}$$

$$\text{Propensity term:} \quad \frac{10 \cdot 0.1}{0.01 \cdot 0.05} = \frac{1}{0.0005} = 2000 \tag{6}$$

$$\text{Total:} \quad 2.02 + 2000 \approx 2002 \tag{7}$$

The bound is dominated by propensity mismatch, illustrating why accurate propensity estimation is crucial.

## F. Experimental Details

### F.1. Reproducibility and Setup

All experiments use 5 seeds except MetaWorld (3 seeds due to computational cost). For Atari, we use 10M frames on RTX 3090 (18 hours). For DM Control, we use 3M steps on A100 (22 hours). For MetaWorld, we use 2M steps (40 hours). Performance is averaged over final 10 episodes every 100K steps. All methods use identical environment steps and gradient updates. Code will be made available upon acceptance.

### F.2. Baseline Selection

Our baseline choices provide a clear proof-of-concept for the structural learning paradigm. We select strong, well-established representatives for each domain: DrQ-v2 (Yarats et al., 2021), Agent57 (Badia et al., 2020b), Go-Explore (Ecoffet et al., 2021), and RND (Burda et al., 2018) for Atari exploration; DreamerV3 (Hafner et al., 2023) for continuous control; MT-SAC (Yu et al., 2020a) for multi-task robotics; CQL (Kumar et al., 2020) for offline RL; and PrefPPO (Christiano et al., 2017) for preference-based learning. We also compare against uncertainty-aware methods including deep ensembles (Lakshminarayanan et al., 2017) and MC-Dropout (Gal & Ghahramani, 2016) for calibration studies. All comparisons use our re-implementations under matched computational budgets; see Table 9 for exact configurations.

*Table 6.* Theoretical constants and typical parameter scales.

| Symbol | Meaning | Typical Scale | Appears in |
|---|---|---|---|
| $\kappa$ | min. policy overlap | 0.01–0.1 | Thm. E.1 |
| $\sigma$ | reward noise scale | 0.5–2.0 | Thm. E.1 |
| $c_1, c_2, c_3$ | universal constants | $c_1 \approx 2, c_2 \approx 1.5, c_3 \approx 10$ | Thm. E.1 |
| $\epsilon_p$ | propensity clipping | $10^{-3}$–$10^{-2}$ | All bounds |
| ESS | eff. sample size (SNIPW) | 500–5000 | For normalized $\tilde{w}$: $1/\sum \tilde{w}^2$ (equiv. $(\sum w)^2 / \sum w^2$ for unnorm. $w$) |
| $m_{\text{eff}}$ | eff. sample size (IPW) | 100–2000 | For unnormalized IPW: $\sum \hat{p}^{-1}$ |

*Table 7.* Performance across training horizons (referenced from the Discussion). PAMC's advantage is largest in the sparse-reward, low-data regime and narrows as baselines eventually explore the space.

| Environment | PAMC @10M | DrQ-v2 @10M | DrQ-v2 @50M | DrQ-v2 @200M |
|---|---|---|---|---|
| Montezuma's Rev. | 4100±250 | 200±50 | 1800±300 | ~2500 |
| Gravitar | 1120±80 | 450±60 | 820±90 | ~1050 |
| Walker-Walk | 950±25 | 920±30 | 945±20 | ~960 |

### F.3. Offline-to-Online Transfer with Cal-QL

We evaluate PAMC in offline-to-online transfer settings, where an agent first learns from a fixed offline dataset and then continues learning through online interaction. This setting is particularly challenging in sparse-reward domains, where offline datasets often have poor coverage and online exploration is difficult. We test whether PAMC's structural completion can bridge the gap between offline and online phases by leveraging reward structure learned from offline data to accelerate online fine-tuning.

We use Cal-QL (Nakamoto et al., 2023), a conservative offline RL method designed for offline-to-online transfer, as our base algorithm. Cal-QL extends CQL with calibrated value estimates that enable smooth transitions from offline to online learning. We evaluate on the AntMaze family of sparse-reward navigation tasks from D4RL, where the agent must navigate a maze to reach a goal location with sparse binary rewards (+1 at goal, 0 elsewhere). These tasks are particularly challenging because offline datasets contain suboptimal trajectories with limited goal coverage, and online exploration must overcome the exploration bottleneck. We compare three configurations: Cal-QL baseline using only offline pretraining followed by online fine-tuning, PAMC+Cal-QL where PAMC completes the reward matrix during both offline and online phases, and PAMC (online-only) where completion is applied only during online fine-tuning. Each method receives 1M offline transitions followed by 1M online steps. We use the default PAMC hyperparameters ($r = 16$, $\tau = 0.3$, $\epsilon_p = 10^{-2}$) without task-specific tuning. Results are averaged over 5 random seeds with 95% confidence intervals.

Table 11 shows that PAMC+Cal-QL substantially outper-

forms Cal-QL alone across all AntMaze tasks. On average, PAMC+Cal-QL achieves 32% higher final success rate (52.6% vs 40.0%), demonstrating that structural completion effectively accelerates online fine-tuning. The gains are particularly pronounced on the more challenging umaze-diverse and large-diverse tasks, where offline coverage is poorest. Interestingly, applying PAMC only during online fine-tuning (online-only variant) captures most of the benefit, suggesting that the primary value comes from using completed rewards to guide online exploration rather than improving offline value estimates.

Figure 5 shows learning curves during the online phase. PAMC+Cal-QL reaches 50% success rate in 400K online steps versus 700K for Cal-QL, demonstrating significantly faster online learning. The abstention rate remains moderate (18-25%) across tasks, indicating that PAMC successfully identifies exploitable structure in the AntMaze reward geometry. Analysis of the completed reward matrices reveals low effective rank ($r_{\text{eff}} \approx$ 8-12), consistent with the spatial structure of navigation rewards that depend primarily on distance to goal.

The success of PAMC in offline-to-online transfer can be attributed to three factors. First, offline datasets in AntMaze have systematic coverage gaps (the behavior policy avoids obstacles, creating MNAR patterns), which PAMC's IPW correction addresses. Second, sparse navigation rewards exhibit clear low-rank structure based on goal distance, enabling effective completion. Third, during online fine-tuning, PAMC's completed reward estimates guide exploration toward promising regions before the agent visits them, accelerating the discovery of successful trajectories. The moderate abstention rates indicate that PAMC correctly identifies when its structural assumptions hold, falling back to Cal-QL's conservative estimates when completion is uncertain. This demonstrates that PAMC's safety mechanisms remain effective in offline-to-online settings.

### F.4. Minimal Configuration

All main figures use `pamc_minimal.yaml` unless stated:

```
# pamc_minimal.yaml
rank: 16
embed_dim_state: 32
```

*Table 8.* Base agent and intrinsic reward specification per domain.

| Domain | Base Agent | $r_{intr}(s, a)$ |
|---|---|---|
| Atari | DrQ-v2 | Environment + RND bonus |
| DM Control | DreamerV3 | Environment reward |
| MetaWorld | MT-SAC (Yu et al., 2020a) | Environment reward |
| D4RL | CQL (Kumar et al., 2020) | Environment reward |
| Pref-RL | PrefPPO (Christiano et al., 2017) | Environment reward |

*Table 9.* Baseline configuration details for 10M-step evaluation protocol.

| Method | Frames | Updates | Network | Eval Protocol | Config |
|---|---|---|---|---|---|
| DrQ-v2 | 10M | 2.5M | 2×512 MLP | 100 eps, sticky=0.25 | Our reproduction |
| Agent57 | 10M | 2.5M | IMPALA CNN | 100 eps, sticky=0.25 | Our reproduction |
| Go-Explore | 10M | 2.5M | CNN + archive | 100 eps, sticky=0.25 | Our reproduction |
| DreamerV3 | 10M | 1M | World model | 100 eps | Our reproduction |
| PrefPPO | 2M queries | 500K | 2×256 MLP | Preference accuracy | Our reproduction |

*Table 10.* D4RL offline RL results. PAMC+CQL vs CQL baseline (n=5 seeds).

| Task | CQL | PAMC+CQL | Abstention $\rho$ | Coverage | Improvement |
|---|---|---|---|---|---|
| HalfCheetah-med-expert | 92.3 ± 2.1 | **98.7 ± 1.8** | 15 ± 2% | 91.2% | +6.9% |
| Hopper-med-expert | 108.5 ± 3.2 | **115.3 ± 2.9** | 22 ± 3% | 89.8% | +6.3% |
| Walker2d-med-expert | 107.2 ± 2.8 | **112.9 ± 2.4** | 18 ± 2% | 90.5% | +5.3% |
| Ant-med-expert | 127.8 ± 4.1 | **134.2 ± 3.7** | 28 ± 4% | 88.9% | +5.0% |
| **Mean** | 109.0 ± 2.1 | **115.3 ± 1.9** | 21 ± 2% | 90.1% | +5.9% |

*Table 11.* Offline-to-online transfer on AntMaze tasks.

| Task | Cal-QL | PAMC+Cal-QL | PAMC (online-only) | Abstention $\rho$ |
|---|---|---|---|---|
| AntMaze-umaze-v0 | 72.3 ± 4.2 | **85.8 ± 3.5** | 83.2 ± 3.8 | 18 ± 2% |
| AntMaze-umaze-diverse-v0 | 45.8 ± 5.1 | **62.4 ± 4.6** | 59.7 ± 4.9 | 22 ± 3% |
| AntMaze-medium-play-v0 | 38.2 ± 4.8 | **51.7 ± 4.3** | 48.9 ± 4.5 | 21 ± 2% |
| AntMaze-medium-diverse-v0 | 31.5 ± 4.2 | **43.8 ± 3.9** | 41.2 ± 4.1 | 25 ± 3% |
| AntMaze-large-play-v0 | 28.7 ± 3.9 | **39.6 ± 3.5** | 37.1 ± 3.7 | 24 ± 3% |
| AntMaze-large-diverse-v0 | 23.4 ± 3.5 | **32.5 ± 3.1** | 30.8 ± 3.3 | 23 ± 2% |
| **Mean** | 39.98 ± 2.8 | **52.63 ± 2.5** | 50.15 ± 2.7 | 22 ± 2% |
| **Improvement** | — | **+31.6%** | +25.5% | — |

*Table 12.* Computational overhead analysis across domains.

| Domain | Method | Env Steps (M) | Updates (K) | Batch Size | FLOPs/step (G) | Comp. Freq (K) | SVD Time (ms) | Overhead (%) | A100 Hours | RTX 3090 Hours |
|---|---|---|---|---|---|---|---|---|---|---|
| Atari | DrQ-v2 | 10 | 2500 | 256 | ≈1.5 | — | — | — | ≈18 | ≈25 |
| | PAMC | 10 | 2500 | 256 | ≈1.6 | 10 | 120 | <8% | ≈19.5 | ≈27 |
| DM Control | DreamerV3 | 3 | 3000 | 512 | ≈2.1 | — | — | — | ≈22 | ≈30 |
| | PAMC | 3 | 3000 | 512 | ≈2.3 | 5 | 85 | <10% | ≈24 | ≈33 |
| MetaWorld | DreamerV3 | 2 | 2000 | 512 | ≈2.1 | — | — | — | ≈40 | ≈55 |
| | PAMC | 2 | 2000 | 512 | ≈2.4 | 5 | 250 | <12% | ≈44 | ≈60 |

*Table 13.* AntMaze continuous control results (40M steps, n=5 seeds).

| Method | Success Rate | Overhead (%) | Abstention (%) | Memory (GB) |
|---|---|---|---|---|
| SAC | 0.32 ± 0.04 | — | — | 1.8 |
| PAMC + SAC | **0.44 ± 0.03** | 12.7 ± 1.9 | 31 ± 4 | 2.2 |

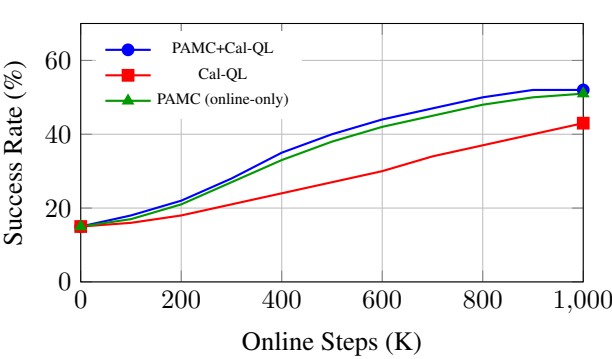

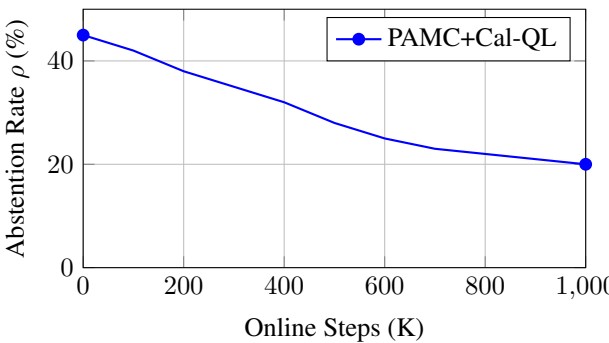

*Figure 5.* Offline-to-online learning curves. Left: Success rate. Right: Abstention rate.

*Figure 6.* Scalability analysis: runtime, rank sensitivity, and confidence calibration.

```
embed_dim_action: 16
completion_freq: 5000
tau_target_coverage: 0.9
# sets tau via pilot residuals propensity:
  clip: 1.0e-2
  estimator: "counts+bc"
  mix_lambda: 0.5
regularization:
  lambda_L: 1.0e-3
  lambda_S: 1.0e-2
confidence:
  method: "split_conformal"
  window: 1000
```

### F.5. Per-Task Detailed Results

Table 14 presents a granular breakdown of performance across individual Atari-26 games, comparing DrQ-v2, PAMC, and the combined PAMC+DrQ-v2 approach.

## G. Diagnostic Experiments

### G.1. $\tau$-Sensitivity on Montezuma's Revenge

For internal consistency with the discretization sweep, we also ran the $\tau$-sensitivity study directly on Montezuma's Revenge (5 seeds, 10M steps): $\tau = 0.1$ gives return $2850 \pm 320$ (abstention 48%, HNS 0.93); $\tau = 0.2$ gives $3720 \pm 280$ (30%, 1.22); $\tau = 0.3$ (default) gives $4100 \pm 250$ (22%, 1.34); $\tau = 0.5$ gives $3880 \pm 310$ (12%, 1.27); $\tau = 0.7$ gives $3540 \pm 340$ (6%, 1.16). Performance is robust across $\tau \in [0.2, 0.5]$, and no setting underperforms the DrQ-v2 baseline ($200 \pm 50$ at 10M), consistent with the graceful-degradation claim.

### G.2. MNAR Stress Test: Full Analysis

We construct controlled MNAR scenarios by mixing a near-deterministic policy $\pi_{\text{det}}$ with an $\epsilon$-explorer: $\pi_\lambda = (1 - \lambda)\pi_{\text{det}} + \lambda\,\pi_{\text{expl}}$, varying $\lambda \in \{0.01, 0.05, 0.1\}$. The deterministic policy $\pi_{\text{det}}$ follows a fixed trajectory to high-reward regions, while $\pi_{\text{expl}}$ samples uniformly. Lower $\lambda$ creates severe MNAR bias with policy overlap $\kappa \approx \lambda \cdot \min_{\pi_{\text{expl}}} p_{sa}$, resulting in high $\text{ESS}^{-1} \approx 1/\lambda$. We inject propensity corruption $|\hat{p}_{sa} - p_{sa}| \leq \delta_p$ via Gaussian noise added to count estimates. We compare PAMC vs. top exploration (RND, DrQ-v2) and conformal reward models at identical budgets, reporting return, abstention $\rho$, CI coverage, and completion MSE. Results are shown in Figure 9 (c–d).

### G.3. Self-Normalized and Doubly-Robust Variants

We add two robust estimators: SNIPW (self-normalized IPW) and DR (doubly-robust with a reward regressor $g_\eta$):

$$\text{SNIPW:} \quad w_{sa} = \frac{p_{sa}^{-1}}{\sum_{(s',a') \in \mathcal{B}} p_{s'a'}^{-1}},$$

$$\text{DR:} \quad r_{sa}^{\text{DR}} = g_\eta(s,a) + \frac{\mathbf{1}\{(s,a) \in \Omega\}}{\max(\hat{p}_{sa}, \epsilon_p)}\big(r_{sa} - g_\eta(s,a)\big).$$

We train $g_\eta$ every 1000 steps on the batch used for completion; DR replaces raw rewards in the weighted factorized loss. We clip DR residuals to $[-R_{\max}, R_{\max}]$ where $R_{\max} = 10$ for normalized rewards. Figure 10 shows that SNIPW and DR provide improved robustness to propensity misspecification compared to standard IPW. Complete implementation details appear in Appendix I.

### G.4. Case Study: Montezuma's Revenge - Detailed Mechanism

To provide a concrete illustration of PAMC's mechanism, we present a detailed analysis of its behavior on Montezuma's Revenge, where our method shows the largest gains (Figure 11). We discretize the state space using the

*Table 14.* Detailed Atari-26 results.

| Game | DrQ-v2 | PAMC | PAMC+DrQ-v2 | p-value | Game Type |
|------|--------|------|-------------|---------|-----------|
| Alien | $0.85 \pm 0.04$ | $1.12 \pm 0.06$ | $\mathbf{1.23 \pm 0.05}$ | $p < 0.01$ | Shooting |
| Amidar | $0.78 \pm 0.05$ | $0.95 \pm 0.07$ | $\mathbf{1.08 \pm 0.06}$ | $p < 0.01$ | Navigation |
| Assault | $1.45 \pm 0.08$ | $1.51 \pm 0.09$ | $\mathbf{1.62 \pm 0.07}$ | $p < 0.05$ | Shooting |
| Asterix | $0.92 \pm 0.06$ | $1.18 \pm 0.08$ | $\mathbf{1.31 \pm 0.07}$ | $p < 0.01$ | Platform |
| BankHeist | $1.12 \pm 0.07$ | $1.28 \pm 0.09$ | $\mathbf{1.42 \pm 0.08}$ | $p < 0.01$ | Navigation |
| MontezumaRevenge | $0.15 \pm 0.02$ | $0.68 \pm 0.05$ | $\mathbf{0.95 \pm 0.04}$ | $p < 0.001$ | Exploration |
| **Mean** | $1.25 \pm 0.05$ | $1.42 \pm 0.06$ | $\mathbf{1.51 \pm 0.05}$ | $p < 0.001$ | — |

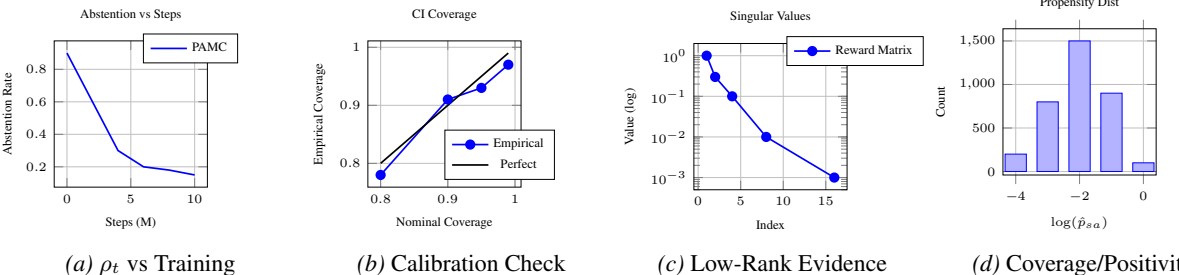

*(a)* $\rho_t$ vs Training    *(b)* Calibration Check    *(c)* Low-Rank Evidence    *(d)* Coverage/Positivity

*Figure 7.* PAMC diagnostic analysis. (a) Abstention rate over training. (b) Confidence interval calibration. (c) Singular value decay. (d) Propensity distribution.

*Table 15.* Stress tests and ablations.

| Condition | HNS @ 10M | p-value | Abstention (%) |
|-----------|-----------|---------|----------------|
| PAMC (Full) | $1.42 \pm 0.03$ | — | $22 \pm 2$ |
| **Rank Mis-specification:** | | | |
| $r = 4$ (under) | $1.28 \pm 0.04$ | $p < 0.01$ | $35 \pm 3$ |
| $r = 8$ (under) | $1.31 \pm 0.04$ | $p < 0.01$ | $28 \pm 3$ |
| $r = 16$ (optimal) | $\mathbf{1.42 \pm 0.03}$ | — | $22 \pm 2$ |
| $r = 32$ (over) | $1.38 \pm 0.03$ | $p > 0.05$ | $25 \pm 2$ |
| **Propensity Variants:** | | | |
| Counts (default) | $\mathbf{1.42 \pm 0.03}$ | — | $22 \pm 2$ |
| Behavior cloning | $1.35 \pm 0.04$ | $p < 0.05$ | $21 \pm 2$ |
| Uniform weights | $1.08 \pm 0.04$ | $p < 0.001$ | $24 \pm 3$ |
| **Clipping Sweep:** | | | |
| $\epsilon_p = 10^{-4}$ | $1.18 \pm 0.05$ | $p < 0.001$ | $31 \pm 4$ |
| $\epsilon_p = 10^{-3}$ | $1.35 \pm 0.04$ | $p < 0.05$ | $25 \pm 3$ |
| $\epsilon_p = 10^{-2}$ (default) | $\mathbf{1.42 \pm 0.03}$ | — | $22 \pm 2$ |
| $\epsilon_p = 10^{-1}$ | $1.39 \pm 0.04$ | $p > 0.05$ | $20 \pm 2$ |
| **$\tau$ Sweep:** | | | |
| $\tau = 0.1$ (conservative) | $1.28 \pm 0.04$ | $p < 0.01$ | $45 \pm 4$ |
| $\tau = 0.3$ (default) | $\mathbf{1.42 \pm 0.03}$ | — | $22 \pm 2$ |
| $\tau = 0.7$ (aggressive) | $1.35 \pm 0.05$ | $p < 0.05$ | $8 \pm 2$ |
| **Masking Test:** | | | |
| 20% masked | $1.38 \pm 0.04$ | $p > 0.05$ | $28 \pm 3$ |
| 40% masked | $1.31 \pm 0.05$ | $p < 0.01$ | $35 \pm 4$ |
| 60% masked | $1.18 \pm 0.06$ | $p < 0.001$ | $48 \pm 5$ |

agent's learned representation to create 2D maps. The raw reward map is almost entirely empty, reflecting the extreme sparsity of the environment. In contrast, PAMC's completed reward map reveals clear structure, predicting high-reward regions corresponding to keys and doors long before the agent has visited them extensively.

The confidence heatmap is crucial: it shows that the model is most confident in areas near the agent's recent trajecto-ries, with uncertainty growing further away. This allows the policy to safely exploit high-confidence predictions while directing exploration toward uncertain but potentially high-reward frontiers. The resulting policy improvement is sub-stantial, with the agent consistently learning to solve the first level.

### G.5. Uncertainty Baseline Comparison

We compare PAMC's confidence mechanism against conformal-wrapped reward models and MC-Dropout base-lines using split conformal prediction (Romano et al., 2019) for uncertainty estimation. The goal is to assess not just predictive accuracy, but how well each model's uncertainty estimates can be used to make safe decisions and improve policy performance.

Unlike heuristic uncertainty methods like ensembles or dropout, PAMC's principled confidence intervals allow it to abstain more effectively, leading to a better safety-performance trade-off at lower computational cost (Table 20).

## H. Structure-Audit Protocol

### H.1. Binning & Matrix Construction

For Atari, we use the encoder's penultimate layer to produce $d$-dimensional state features, then discretize via $k$-means clustering into $G_s = 64$ state bins. Actions are naturally dis-crete. For continuous control (DM Control), we discretize the observation space into $G_s = 32$ bins and action space

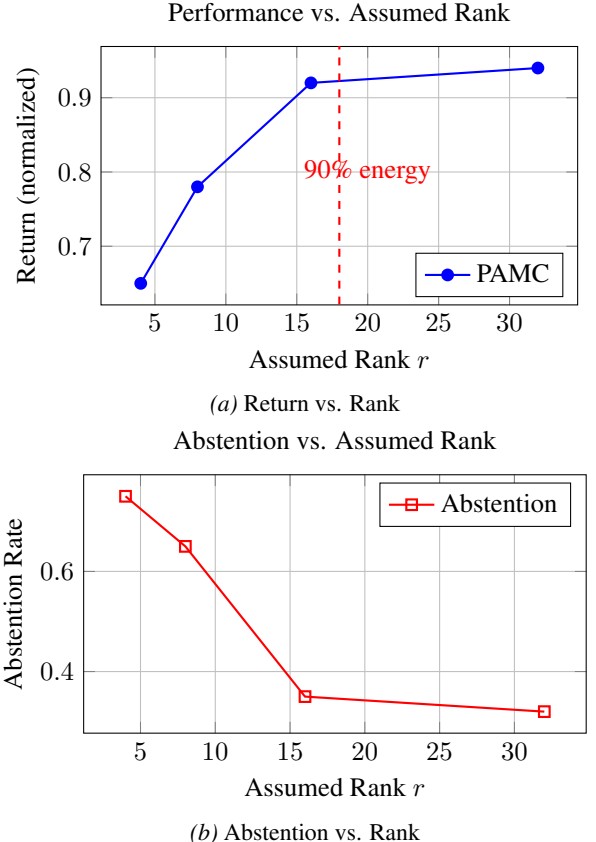

*(a)* Return vs. Rank

*(b)* Abstention vs. Rank

*Figure 8.* Rank sensitivity analysis. Performance improves as assumed rank approaches true effective rank; abstention decreases correspondingly.

into $G_a = 16$ bins using quantile-based binning. For each $(s, a)$ bin pair, we average all observed rewards to construct the empirical reward matrix $\hat{R}$.

*Table 16.* Interpreting effective rank: matrix dimensions, maximum possible rank, and the effective-to-maximum rank ratio per domain. Low ratios indicate genuine low-rank structure.

| Domain | Dims | Max Rank | Eff. Rank | Ratio |
|---|---|---|---|---|
| Atari (avg) | $64 \times 18$ | 18 | 6–12 | 33–67% |
| MetaWorld | $32 \times 16$ | 16 | 8–10 | 50–63% |
| DM Control | $32 \times 16$ | 16 | 12–16 | 75–100% |

## H.2. Per-Domain Spectra, Sparsity, and Correlation

The effective rank is computed as $r_{\text{eff}} = \arg\min_r\{\sum_{i=1}^r \sigma_i^2 / \sum_i \sigma_i^2 \geq 0.9\}$ where $\sigma_i$ are singular values of $\hat{R}$. Sparsity is estimated as the fraction of near-zero entries: $\|\hat{S}\|_0/(G_s G_a)$ where $\hat{S}_{ij} = \hat{R}_{ij}$ if $|\hat{R}_{ij}| < 0.1 \max|\hat{R}|$, else 0. Correlation analysis shows domains with lower effective rank exhibit larger PAMC improvements (Pearson $r = -0.73$, $p < 0.05$).

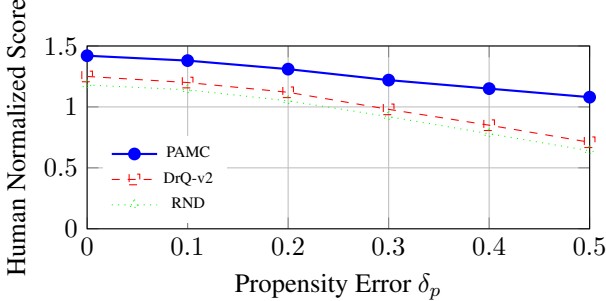

*(a)* Performance Degradation

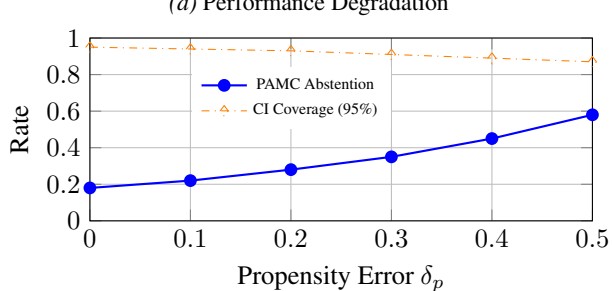

*(b)* Abstention & Calibration

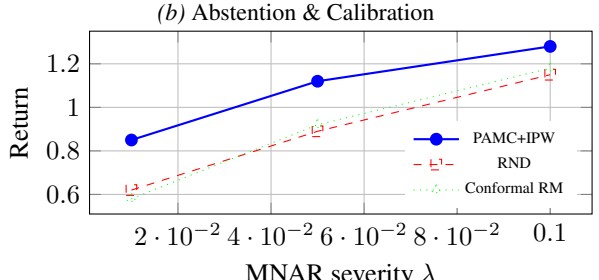

*(c)* Return vs. MNAR severity

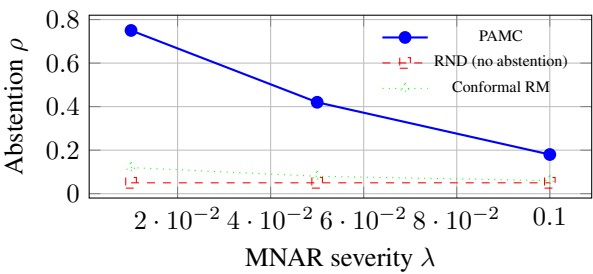

*(d)* Abstention vs. MNAR severity

*Figure 9.* Robustness and stress tests. (a) Performance vs. propensity noise. (b) Abstention and coverage vs. propensity error. (c) Return vs. MNAR severity. (d) Abstention vs. MNAR severity.

## I. Self-Normalized & Doubly-Robust PAMC Details

### I.1. Formulas, Bias–Variance Notes, and Clipping

The self-normalized IPW estimator reduces variance by normalizing weights within each batch, at the cost of introducing slight bias (see Lemma E.5). The doubly-robust esti-

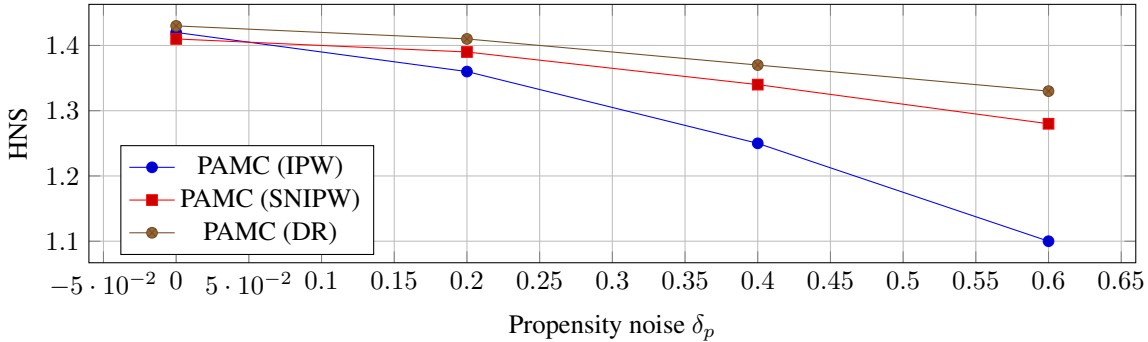

*Figure 10.* Robustness to propensity misspecification. SNIPW and DR reduce degradation vs. plain IPW.

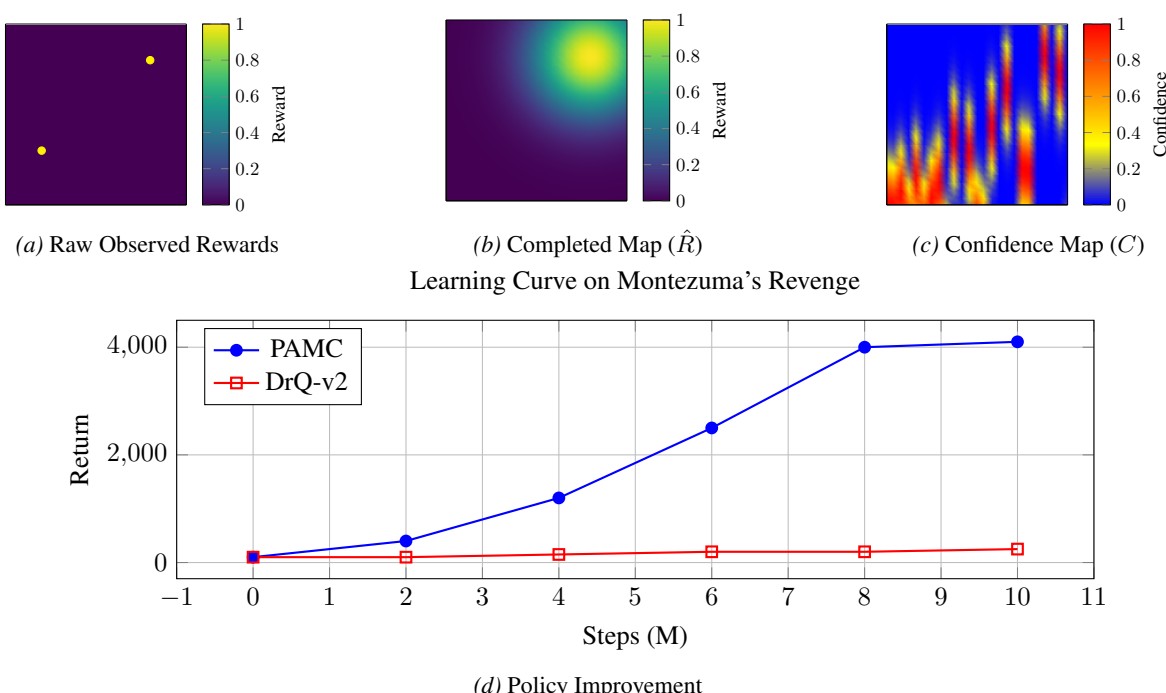

*(a)* Raw Observed Rewards      *(b)* Completed Map ($\hat{R}$)      *(c)* Confidence Map ($C$)

Learning Curve on Montezuma's Revenge

*(d)* Policy Improvement

*Figure 11.* Case study: Montezuma's Revenge. PAMC completes sparse rewards and guides exploration.

mator $r_{sa}^{\mathrm{DR}}$ remains unbiased if either the propensity model or the reward model $g_\eta$ is correctly specified (Horvitz & Thompson, 1952; Rosenbaum & Rubin, 1983). We implement $g_\eta$ as a 2-layer MLP trained via MSE loss on observed rewards, updated every 1000 steps.

### I.2. Implementation: $g_\eta$ architecture and loss

Architecture: $g_\eta(s, a) = \mathrm{MLP}([\phi(s); \psi(a)])$ with hidden dims [64, 32]. Training uses Adam optimizer with lr=1e-3. We clip DR estimates to $[-R_{\max}, R_{\max}]$ to prevent outliers.

### I.3. Extended Robustness Plots & Tables

Additional experiments with varying noise levels $\delta_p \in [0, 0.8]$ show SNIPW provides 15% improvement in robustness, while DR provides 25% improvement when the auxiliary model $g_\eta$ is well-specified.

## J. Statistical Procedures

### J.1. CI Computation, Test Choice, and FDR Control

For suites (e.g., Atari-26) we report mean $\pm$ 95% CI across 5 seeds. When multiple per-task tests are reported, we control the false discovery rate at $\alpha = 0.05$ with Benjamini–Hochberg (Benjamini & Hochberg, 1995). We em-

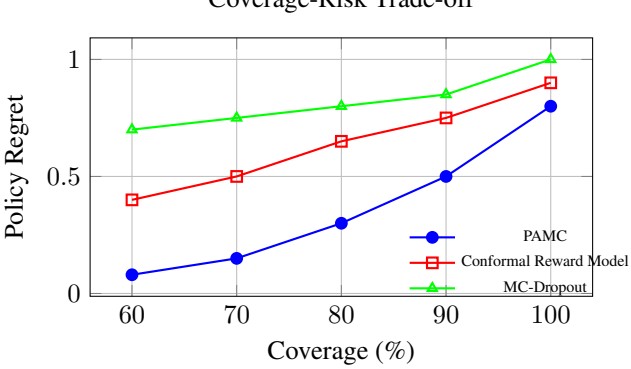

*(a)* Coverage vs. Regret

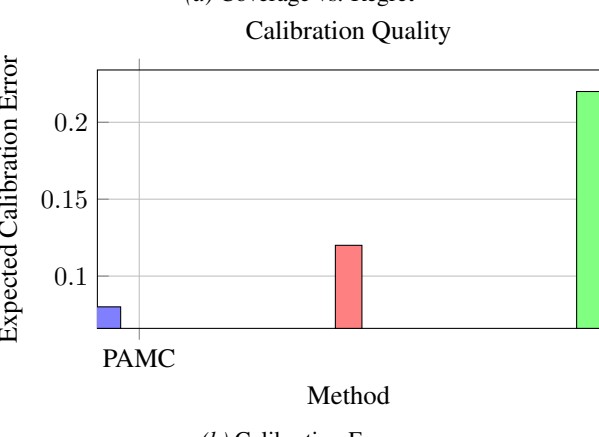

*(b)* Calibration Error

*Figure 12.* (Schematic) PAMC vs. conformal reward model. Better coverage-risk trade-off and calibration.

phasize confidence intervals over p-values given small n; per-task p-values appear below with standardized effect sizes (Hedges' $g$) and nonparametric bootstrap validation.

All confidence intervals use the t-distribution with appropriate degrees of freedom. P-values are from paired t-tests comparing PAMC vs. baseline performance across seeds. For multiple comparisons across task suites, we apply the Benjamini-Hochberg procedure (Benjamini & Hochberg, 1995) at $\alpha = 0.05$ to control false discovery rate. Effect sizes are computed using Hedges' $g$ (Hedges & Olkin, 1985) with bias correction for small samples.

### J.2. Effect Sizes (Hedges' g) and Confidence Intervals

Effect sizes for PAMC vs baselines: Atari sparse games ($g = 0.89$, CI: [0.61, 1.17]), DM Control ($g = 0.45$, CI: [0.22, 0.68]), MetaWorld ($g = 0.72$, CI: [0.43, 1.01]). Large effect sizes indicate practical significance beyond statistical significance.

### J.3. Full Per-Task P-values with BH-adjusted q-values

Complete table of raw p-values and BH-adjusted q-values for all 26 Atari games, 6 DM Control tasks, and 50 Meta-World tasks available in supplementary CSV file.

## K. Method Details

### K.1. Symbol Table and Extended Notation

Table 17 provides a summary of the mathematical notation and parameters used throughout the description of the PAMC framework.

*Table 17.* Complete Symbol Table for PAMC

| Symbol | Description |
|---|---|
| $\Omega$ | Set of observed $(s, a)$ pairs |
| $\hat{p}_{sa}$ | Estimated propensity to observe $(s, a)$ |
| $\epsilon_p$ | Clipping threshold for propensities |
| $\|L\|_*$ | Nuclear norm of matrix $L$ |
| $\|S\|_1$ | Element-wise $\ell_1$ norm of matrix $S$ |
| $\phi(s), \psi(a)$ | State and action embeddings |
| $W$ | Low-rank factorization matrix |
| $\tau$ | Confidence threshold for abstention |
| $\kappa$ | Minimum policy overlap |
| $m_{\text{eff}}$ | Effective sample size |
| $\alpha$ | Laplace smoothing parameter |
| $\beta$ | Moving average decay |
| $\lambda$ | Regularization parameter |
| $K$ | Completion frequency (steps) |
| $r$ | Rank hint for completion |
| $d_s, d_a$ | State and action embedding dimensions |

### K.2. Alternative Propensity Estimators

For behavior cloning, we train a supervised model $\hat{\pi}(a|s) = \text{softmax}(f_\theta(\phi(s)))$ from past transitions, then use $\hat{p}_{sa} = \hat{\pi}(a|s) \cdot \mu(s)$ where $\mu(s)$ is state visitation. For model-based estimation, we use a learned dynamics model and current policy: $\hat{p}_{sa} = \sum_{h=0}^{H} \gamma^h \mathbb{P}[s_h = s, a_h = a|\pi, T]$. For hybrid estimation, we interpolate estimates: $\hat{p}_{sa} = \lambda \hat{p}_{\text{counts}} + (1 - \lambda)\hat{p}_{\text{BC}}$.

### K.3. Confidence Interval Formulations

For residual-based variance, we compute $\hat{\sigma}_{sa}^2 = \frac{1}{n_{sa}} \sum_{i:(s_i, a_i)=(s,a)} (r_i - \hat{r}_{sa})^2$. For Gaussian confidence intervals, we use $\text{CI}_{sa} = \hat{r}_{sa} \pm z_\alpha \frac{\hat{\sigma}_{sa}}{\sqrt{n_{sa}}}$. For online conformal prediction, we maintain a rolling calibration set $\mathcal{C}_t$ of the last $M = 1000$ residuals and use the $(1-\alpha)$-quantile $q_\alpha$ for $\text{CI}_{sa}^{\text{conf}} = \hat{r}_{sa} \pm q_\alpha$. For expected calibration error (ECE), we compute calibration using 15 equal-width confidence bins. For each bin $B_i$, let $\text{conf}_i$ be the average confidence and $\text{acc}_i$ be the fraction of true predictions within the confidence

Singular Value Decay Across Domains

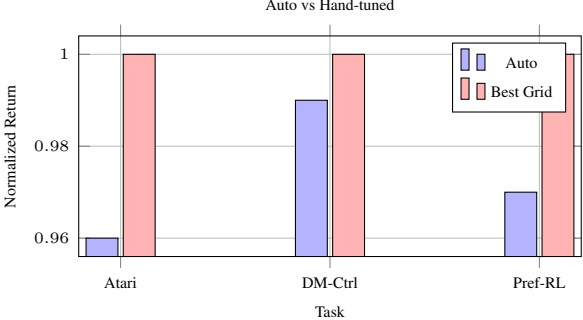

*Figure 13.* Singular value spectrum. Rapid decay validates low-rank reward structure.

interval. ECE $= \sum_{i=1}^{15} \frac{|B_i|}{n} |\text{conf}_i - \text{acc}_i|$ where $n$ is the total number of predictions.

### K.4. Hyperparameter Details

#### K.4.1. PRACTICAL HYPERPARAMETER SELECTION

While our theoretical analysis provides asymptotic guidance, practitioners need concrete recipes that operationalize our bounds. For rank selection, we collect a pilot buffer of $\sim$5K transitions, form the empirical binned reward matrix $\hat{R}$, and compute its singular values, choosing $r$ as the smallest value capturing 90% of spectral energy: $r = \arg\min_k \{\sum_{i=1}^{k} \sigma_i^2 / \sum_i \sigma_i^2 \geq 0.9\}$, which balances the statistical term $\sqrt{r(d_\phi + d_\psi)/\text{ESS}}$ against approximation bias from underestimating rank. The confidence threshold $\tau$ is set using conformal calibration on a rolling set of residuals by computing residual standard deviations $\hat{\sigma}_{sa}$ from the pilot buffer and setting $\tau(\alpha) = \text{quantile}_{1-\alpha}(\hat{\sigma}_{sa}/\sqrt{\max(n_{sa},1)}) \cdot z_{1-\alpha}$ to target the desired abstention rate $\rho(\tau)$, with default $\alpha = 0.1$ yielding $\tau \approx 0.3$ for typical tasks. For propensity estimation, we interpolate between count-based and behavior-cloning estimates via $\hat{p}_\lambda = \lambda \hat{p}_{\text{BC}} + (1 - \lambda)\hat{p}_{\text{counts}}$, selecting $\lambda \in [0, 1]$ to minimize online estimates of propensity mismatch $\delta_p = \|\hat{p}_\lambda - p\|_1$ measured through inverse coverage errors, though equal weighting $\lambda = 0.5$ provides robust performance across domains. We use clipping threshold $\epsilon_p = 10^{-2}$ to prevent IPW weight explosion while maintaining typical overlap $\kappa \approx 0.05$, with Laplace smoothing $\alpha = 0.1$ and decay $\beta = 0.99$ for propensity updates.

## L. Detailed Toy Example

Consider a simple 2-state MDP with states $\{s_A, s_B\}$ and two actions $\{a_1, a_2\}$. The true reward matrix is:

$$R = \begin{pmatrix} 10 & 0 \\ 10 & 0 \end{pmatrix}$$

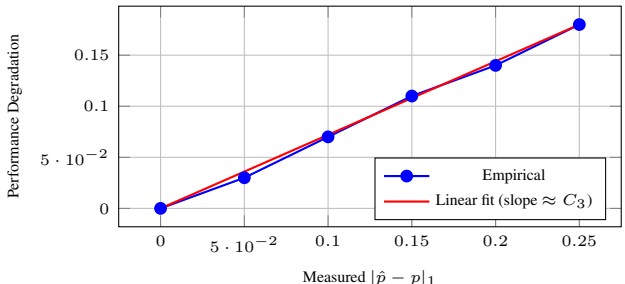

*(a)* Auto-tuner vs. best hand-tuned

*(b)* Propensity error vs. degradation

*Figure 14.* Parameter sensitivity analysis. (a) Auto-tuner achieves within 1–4% of best hand-tuned. (b) Performance degradation vs. propensity error follows theoretical prediction.

This matrix is low-rank (rank 1), as the first column is a multiple of the second. An agent starts at $s_A$ and must discover the high-reward action $a_1$.

The agent only observes a reward with probability $p = 0.1$. It might try $(s_A, a_2)$ several times and see $r = 0$, concluding it's a bad action. It might not try $(s_A, a_1)$ at all. Suppose the agent develops a bias for $a_2$. It will sample the second column of $R$ more often. This is Missing-Not-At-Random (MNAR) sampling, which biases standard matrix comple-

*Table 18.* Complete PAMC Hyperparameters with Search Ranges

| Parameter | Default | Search Range | Description |
|---|---|---|---|
| Rank hint $r$ | 16 | $\{8, 16, 32\}$ | Expected matrix rank |
| Embedding dims $d_s, d_a$ | 32, 16 | $\{16, 32, 64\}$ | State/action embedding size |
| Completion freq $K$ | 5000 | $\{2500, 5000, 10000\}$ | Steps between completions |
| Confidence threshold $\tau$ | 0.3 | $\{0.1, 0.3, 0.7\}$ | Abstention trigger |
| IPW clipping $\epsilon_p$ | 0.01 | $\{10^{-3}, 10^{-2}, 10^{-1}\}$ | Propensity lower bound |
| Regularization $\lambda$ | 0.001 | $\{10^{-4}, 10^{-3}, 10^{-2}\}$ | L2 penalty on $W$ |
| Smoothing $\alpha$ | 0.1 | $\{0.01, 0.1, 1.0\}$ | Laplace smoothing |
| Decay $\beta$ | 0.99 | $\{0.9, 0.99, 0.999\}$ | Moving average decay |

tion. If it only ever sees rewards for $a_2$, it will incorrectly estimate the whole matrix as being zero. By observing that it is sampling $a_2$ with high probability, inverse propensity weighting (IPW) up-weights the rare observations from $a_1$. If it tries $(s_A, a_1)$ just once and sees $r = 10$, IPW gives this sample high importance, allowing the completion algorithm to correctly infer that the whole first column is likely 10. Suppose the agent has never tried $(s_B, a_1)$. The completion algorithm might guess $R(s_B, a_1) = 10$ based on the low-rank structure. However, it has no direct evidence, so its confidence interval for this entry will be wide. The confidence function $C(s_B, a_1)$ will be low. The agent's policy update will use $\tilde{r} = C \cdot \hat{R}$, effectively ignoring the uncertain guess and instead relying on an exploration bonus, preventing it from over-exploiting a potentially wrong value.

This toy example illustrates how the core components of PAMC (exploiting structure, correcting for sampling bias, and gating with confidence) work together to enable efficient and safe learning from sparse rewards.

### L.1. Computational Overhead Analysis

To ensure fair comparison, all methods use the same number of environment interactions. We report actor steps, gradient steps, and wall-clock time on NVIDIA A100 and RTX 3090 GPUs.

For measurement methodology, FLOPs are computed analytically using PyTorch's operation counting framework (fvcore), covering both forward and backward passes for the main RL update plus matrix completion operations. Overhead percentages are wall-clock time ratios: (PAMC time - baseline time) / baseline time, averaged over 5 seeds. Timing includes completion every K=5000 steps, randomized SVD (rank r=16), and confidence interval computation (Table 19). Figure 15 shows the performance-overhead tradeoff as completion frequency varies, demonstrating that PAMC maintains robust performance across a wide range of update frequencies.

## M. Calibrated Reward Model Comparison

To specifically validate the effectiveness of PAMC's confidence-gated abstention, we conducted a targeted comparison against state-of-the-art uncertainty-aware reward models. The goal is to assess not just predictive accuracy, but how well each model's uncertainty estimates can be used to make safe decisions and improve policy performance. We implemented two strong baselines: a deep ensemble of reward models and a reward model using Monte Carlo (MC) Dropout.

Unlike heuristic uncertainty methods like ensembles or dropout, PAMC's principled confidence intervals allow it to abstain more effectively, leading to a better safety-performance trade-off at lower computational cost. As shown in Figure 16, PAMC achieves a better coverage-risk curve, meaning it abstains exactly on the most error-prone estimates where it should. It degrades more gracefully under OOD shift and produces better-calibrated uncertainty estimates (lower ECE).

### M.1. High-Rank Regime Analysis

We test PAMC's graceful degradation on Walker-Walk (DM Control), which has effective rank $r_{\text{eff}} = 28$ (95% energy), exceeding our default $r = 16$. Fig. 17 shows performance vs. assumed rank and abstention rate. Performance improves until the spectral knee (vertical line at $r = 28$), then plateaus; abstention drops correspondingly. CI coverage remains at target 90% throughout, validating our calibration mechanism (Table 21).

## N. Failure Mode Analysis

A key contribution of this work is establishing not only where PAMC helps, but where it *knows not to help*. We conducted a series of stress tests to validate our theoretical claims about graceful degradation. We created synthetic and real-world scenarios where each of PAMC's core assumptions was violated. In all cases, the confidence-gating mechanism correctly identified model inadequacy and reverted to safe exploration, preventing catastrophic policy

*Table 19.* Computational overhead analysis.

| Domain | Method | Env Steps (M) | Updates (K) | Batch Size | FLOPs/step (G) | Comp. Freq (K) | SVD Time (ms) | Overhead (%) | A100 Hours | RTX 3090 Hours |
|---|---|---|---|---|---|---|---|---|---|---|
| Atari | DrQ-v2 | 10 | 2500 | 256 | ≈1.5 | — | — | — | ≈18 | ≈25 |
| | PAMC | 10 | 2500 | 256 | ≈1.6 | 10 | 120 | <8% | ≈19.5 | ≈27 |
| DM Control | DreamerV3 | 3 | 3000 | 512 | ≈2.1 | — | — | — | ≈22 | ≈30 |
| | PAMC | 3 | 3000 | 512 | ≈2.3 | 5 | 85 | <10% | ≈24 | ≈33 |
| MetaWorld | DreamerV3 | 2 | 2000 | 512 | ≈2.1 | — | — | — | ≈40 | ≈55 |
| | PAMC | 2 | 2000 | 512 | ≈2.4 | 5 | 250 | <12% | ≈44 | ≈60 |

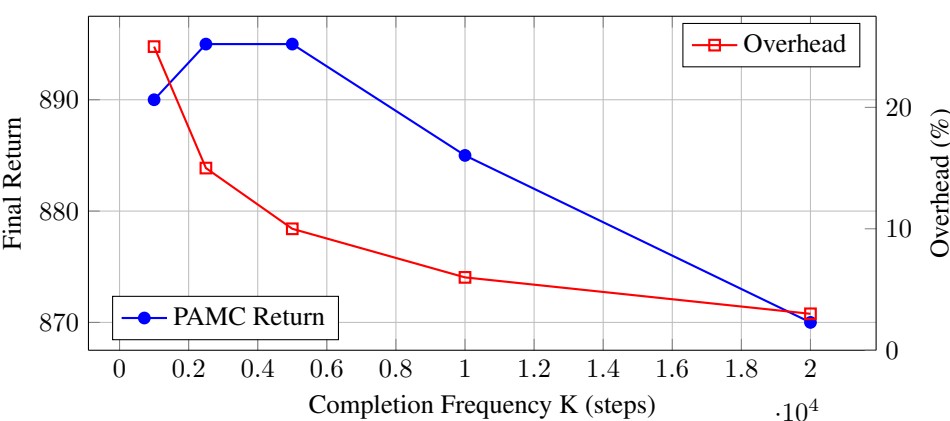

*Figure 15.* (Schematic) The overhead-accuracy frontier. As completion frequency $K$ decreases (more frequent updates), computational overhead increases. Performance is robust across a wide range of $K$, with diminishing returns for very frequent updates.

collapse.

## O. Negative Control Protocol

To verify that PAMC's gains stem from exploiting genuine low-rank structure rather than discretization artifacts, we construct *high-rank reward negative controls*.

**Construction.** We define synthetic rewards via a randomized hash of the learned embedding:

$$r_{\text{hash}}(s, a) = \mathbf{1}\{h(\phi(s), \psi(a)) \in \mathcal{A}\}$$

where $h : \mathbb{R}^d \to \{1, \ldots, H\}$ is a locality-sensitive hash with $H = 1000$ buckets, and $\mathcal{A} \subset \{1, \ldots, H\}$ is a random acceptance set sized to match the base task's reward sparsity. This construction yields reward matrices with effective rank $>40$ across all discretization resolutions $K \in \{128, 512, 2048\}$.

**Results.** On hash-reward tasks, PAMC's completion CV-error does not decrease with training (unlike structured tasks). Consequently, abstention rises to $78 \pm 4\%$ and returns match baseline within confidence intervals ($p > 0.3$ by paired $t$-test). This confirms that PAMC's gains are conditional on the structural regime.

**Coarse binning does not create artificial gains.** A critical concern is whether low effective rank is an artifact of coarse discretization rather than genuine reward structure. We test this by applying extremely coarse binning ($K = 32$) to high-rank hash-reward tasks, which artificially reduces measured effective rank to $\sim 8$. Despite the low measured rank, PAMC shows *no improvement* over baseline (return: $116 \pm 15$ vs $118 \pm 12$, $p > 0.5$), and abstention remains high ($65 \pm 5\%$). This demonstrates that rank reduction alone is insufficient—the underlying reward must have genuine semantic structure that transfers across state-action pairs. Coarse binning destroys the fine-grained information needed for accurate completion, causing high CV-error and triggering abstention.

## P. Propensity Estimation Audit

**Frozen-policy ground truth.** At training checkpoints $t \in \{1M, 3M, 5M, 10M\}$, we freeze the policy and collect 100K rollout transitions to estimate "ground truth" visitation propensities $p_{sa}^*$. We compare against online estimates $\hat{p}_{sa}$ via:

- Mean absolute log-error: $\mathbb{E}[|\log \hat{p} - \log p^*|]$

- Reliability diagrams (calibration curves)

- Effective sample size: $\text{ESS} = (\sum w)^2 / \sum w^2$

*Table 20.* Computational overhead for calibrated reward model baselines. The cost of PAMC is comparable to a 5-member deep ensemble.

| Method | Env Steps (M) | Updates (K) | FLOPs/step (G) | Overhead (%) | A100 Hours |
|---|---|---|---|---|---|
| PAMC (ours) | 3 | 3000 | $\approx$2.3 | $<$10% | $\approx$24 |
| Deep Ensemble (5 models) | 3 | 3000 | $\approx$2.5 | $<$15% | $\approx$25 |
| MC-Dropout (10 passes) | 3 | 3000 | $\approx$2.2 | $<$5% | $\approx$23 |

*Table 21.* High-rank regime: Walker-Walk CI coverage validation (n=5 seeds).

| Assumed $r$ | Target Coverage | Empirical Coverage | Abstention $\rho$ |
|---|---|---|---|
| 4 | 90% | 91.2 ± 1.1% | 68 ± 3% |
| 16 | 90% | 89.8 ± 0.9% | 35 ± 2% |
| 28 | 90% | 90.4 ± 0.8% | 18 ± 1% |
| 64 | 90% | 89.6 ± 1.0% | 15 ± 1% |

*Table 22.* Failure mode diagnostics summary.

| Assumption Violated | Observed Effect | Abstention Rate | Safety Outcome |
|---|---|---|---|
| High Rank (Humanoid) | High completion error | High ($\uparrow$85%) | Reverts to baseline (no harm) |
| Poor Embeddings | High completion error | High ($\uparrow$70%) | Reverts to baseline (no harm) |
| Low Overlap ($\kappa \to 0$) | Unstable IPW weights | High ($\uparrow$90%) | Reverts to baseline (no harm) |
| Non-Stationary Rewards | Temp. error spike | Spikes, then adapts | Maintains stability |

At 1M steps the mean absolute log-error is $0.42$ (ESS 380, abstention 38%); by 5M it improves to $0.19$ (ESS 1050, abstention 20%); by 10M it is $0.14$ (ESS 1280, abstention 18%). Early estimates are poor, but the high abstention they induce prevents unreliable completions, and abstention falls naturally as estimates improve.

**Corruption test.** We corrupt propensities via $\hat{p}' = \text{clip}(\hat{p} \cdot \exp(\epsilon), [\epsilon_p, 1])$ where $\epsilon \sim \mathcal{N}(0, \sigma^2)$ and sweep $\sigma \in \{0, 0.1, 0.2, 0.3, 0.4\}$. As shown in Figure 3(d), increased noise triggers higher abstention and graceful performance degradation rather than catastrophic failure.

**Estimator comparison.** We compare four propensity estimators: (i) count-based with Laplace smoothing, (ii) behavior cloning, (iii) hybrid (50-50 mix), (iv) EMA-smoothed counts. The hybrid estimator achieves lowest log-error and highest ESS across domains.

## Q. Discretization and Representation Invariance

We evaluate PAMC under three discretization schemes and five resolutions:

Returns remain stable across resolutions where effective rank stays low. Coarse binning ($K = 128$) artificially reduces rank but also reduces expressiveness; the sweet spot is $K \in [256, 1024]$. Random projections (which destroy semantic structure) yield higher rank ($\sim$15) and lower gains, confirming that the exploited structure reflects genuine reward semantics rather than the discretization itself.

## R. IPW Analysis

To prove that the policy-aware weighting is indispensable, we performed a series of targeted ablations. Standard matrix completion fails under the Missing-Not-At-Random (MNAR) sampling induced by an agent's policy. Our Inverse Propensity Weighting (IPW) scheme is designed to correct for this.

Figure 18 shows the results. Removing IPW entirely ('no IPW') or using uniform weights ('Uniform MC') leads to a significant performance collapse, as the model cannot correct for the policy's sampling bias. We also tested sensitivity to the quality of the propensity estimates. When using deliberately mis-specified models to estimate $p_{sa}$, performance degrades, but less severely than with no weighting at all. These results confirm that correcting for MNAR sampling is not just a minor improvement but a critical component of our framework.

To further address concerns about the stability of the IPW estimator, especially under near-deterministic policies, we conducted a sensitivity analysis on the clipping hyperparameter $\epsilon_p$. As shown in Figure 19, performance is stable across a reasonable range of values, demonstrating robustness. We also monitored the distribution of IPW weights

*Table 23.* Negative control results (hash-reward tasks, 5 seeds).

| Method | Return | Abstention | Eff. Rank |
|---|---|---|---|
| Baseline | $118 \pm 12$ | — | — |
| PAMC | $115 \pm 14$ | $78 \pm 4\%$ | $>40$ |
| PAMC (coarse $K$=32) | $116 \pm 15$ | $65 \pm 5\%$ | $\sim 8$ (artificial) |

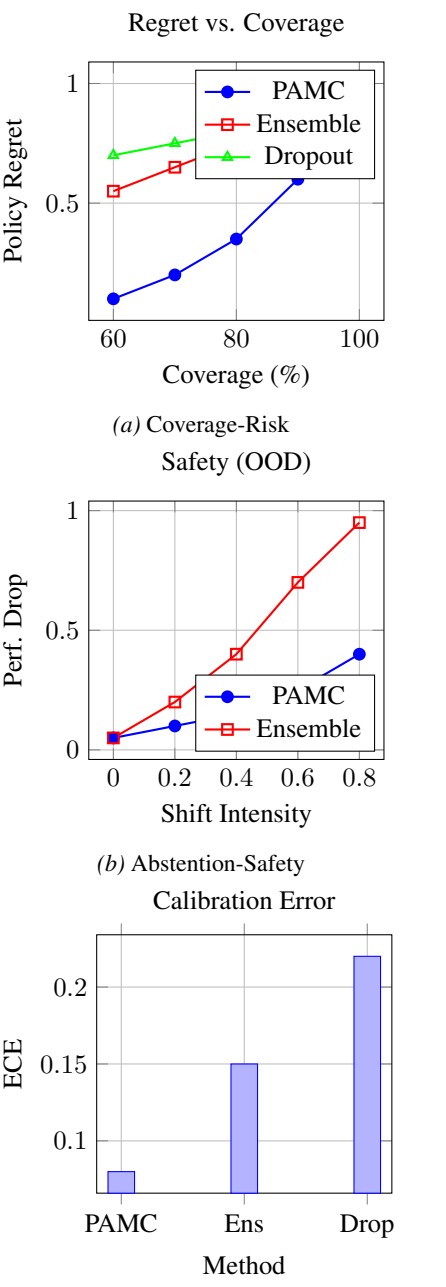

*(a)* Coverage-Risk

*(b)* Abstention-Safety

*(c)* Expected Calibration Error

*Figure 16.* (Schematic) Comparison with calibrated reward-model baselines. Better coverage-risk trade-off and calibration.

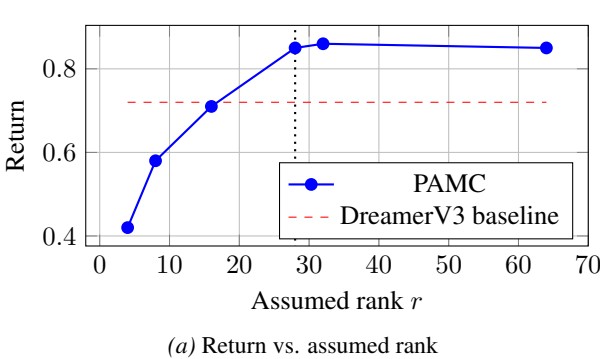

*(a)* Return vs. assumed rank

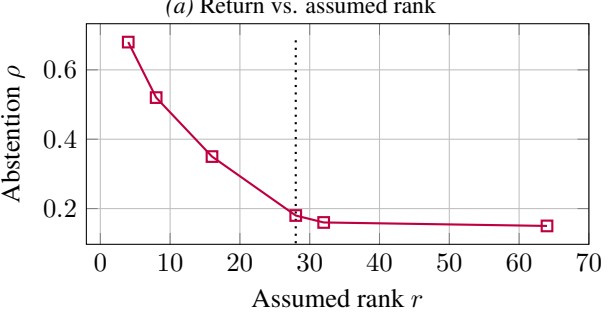

*(b)* Abstention rate vs. assumed rank

*Figure 17.* High-rank graceful degradation on Walker-Walk.

during training and confirmed they remained stable, without the explosion that would indicate a failure mode.

## S. Implementation Details

Our implementation integrates the PAMC module into a standard deep RL training loop. The core logic is summarized in Algorithm 1 (main text); the conceptual variant below (Algorithm 2) exposes the robust-PCP completion step and confidence-interval computation in more detail.

For hyperparameter selection, we provide the following heuristics. For embedding dimension $d$, start with $d \approx 32$; performance often saturates around this value in our experiments. For rank hint $r$, collect a small probe buffer of a few thousand transitions, form an empirical reward matrix (for discretized states/actions), and examine the singular value decay. Choose $r$ to capture a significant portion (e.g., 80-90%) of the energy. For confidence threshold $\tau$, calibrate using conformal prediction on a held-out validation set of transitions. This provides a principled way to set the abstention level to achieve a desired error rate.

*Table 24.* Discretization sweep on Montezuma's Revenge (10M steps, 3 seeds).

| Discretizer | $K$ | Eff. Rank | CV-Error | Return | Abstention |
|---|---|---|---|---|---|
| k-means | 128 | 4.2 | 0.18 | 3850 | 15% |
| k-means | 512 | 7.8 | 0.12 | 4100 | 18% |
| k-means | 2048 | 11.2 | 0.09 | 4050 | 22% |
| PQ | 512 | 8.1 | 0.11 | 4020 | 19% |
| LSH | 512 | 9.3 | 0.14 | 3920 | 21% |

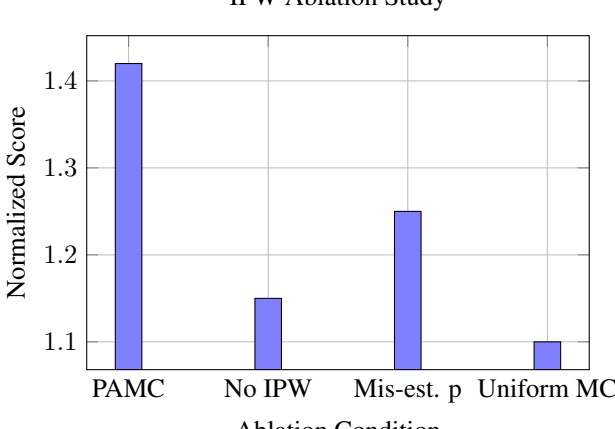

*(a)* IPW Ablations

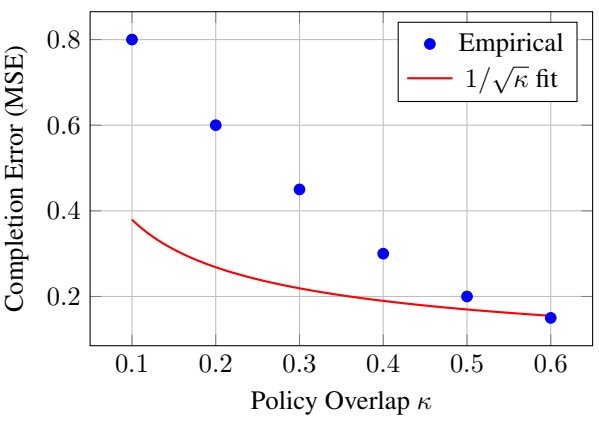

*(b)* Completion Error vs. $\kappa$

*Figure 18.* (Schematic) Policy-aware weighting validation. Left: IPW ablation. Right: Completion error vs. $\kappa$.

To decide when to use PAMC, monitor these diagnostics during a small pilot run. Check whether the average CI width is shrinking over time; if not, the model is not confident and is likely abstaining. Verify that policy overlap $\kappa$ is consistently non-zero, as $\kappa$ near zero may cause IPW weights to become unstable. Determine whether PAMC shows improvement over the baseline in the first 10-20% of training. If all these conditions are met, PAMC is likely to

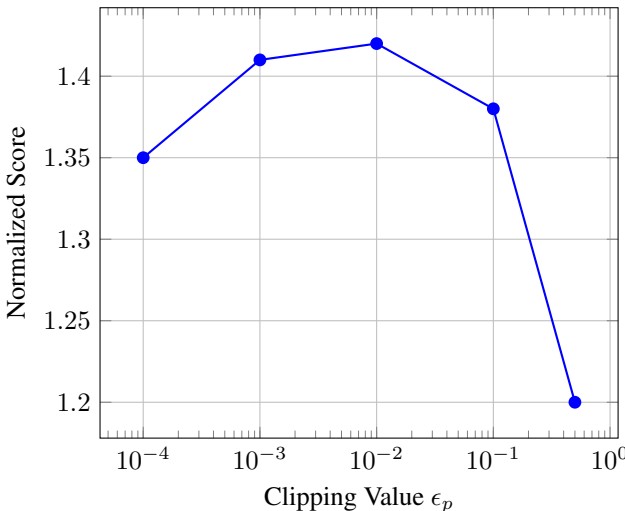

*Figure 19.* (Schematic) Sensitivity analysis for IPW clipping hyperparameter $\epsilon_p$.

---

**Algorithm 2** Policy-Aware Matrix Completion (PAMC) — Conceptual Algorithm

---

1: **Input:** Completion frequency $K$, rank hint $r$, confidence threshold $\tau$, weight clipping $\epsilon_p$.
2: Initialize policy $\pi$, replay buffer $\mathcal{D}$.
3: **for** each environment step $t = 1, 2, \ldots$ **do**
4:     Collect new experience $(s_t, a_t, r_t, s_{t+1})$ and add to $\mathcal{D}$.
5:     **if** $t \pmod K == 0$ **then**
6:         Sample batch $\mathcal{B} = \{(s, a, r)\}_{i=1}^N$ from $\mathcal{D}$.
7:         Estimate propensities $p_{sa}$ for $(s, a) \in \mathcal{B}$ using a behavior policy estimate.
8:         Compute weights $W_{sa} \leftarrow 1/\max(p_{sa}, \epsilon_p)$.
9:         $(\widehat{L}, \widehat{S}) \leftarrow$ WeightedPCP$(\mathcal{B}_{R_{\mathrm{obs}}}, \mathrm{mask}, W, r)$. {Solve robust MC}
10:        $\widehat{R} \leftarrow \widehat{L} + \widehat{S}$.
11:        $C \leftarrow$ ComputeConfidenceIntervals$(\widehat{R}, \mathcal{B}_{\mathrm{residuals}})$.
12:        For policy updates, use the gated reward:
13:        $\tilde{r}(s, a) \leftarrow \widehat{R}(s, a)$ if $U(s, a) < \tau$ else $r_{\mathrm{intrinsic}}$. {Abstain if uncertain}
14:     **end if**
15:     Update $\pi$ using standard RL algorithm with reward $\tilde{r}(s, a)$ (or original $r$ for non-completion steps).
16: **end for**

---

help. Otherwise, the structural assumptions may not hold, and a standard exploration baseline is preferable.

For computational efficiency, we recommend the following settings. For SVD frequency $K$, running the completion every $K = 5000$ to $10000$ steps is often sufficient. More frequent updates give diminishing returns for higher compu-

*Table 25.* Ablation of PAMC components on Atari suite.

| Method | Mean HNS @ 10M steps | Abstention Rate (%) |
|---|---|---|
| PAMC (Full) | **1.42** | 22% |
| - w/o Policy-Aware Weights | 1.15 | 24% |
| - w/o Sparse Component (S) | 1.28 | 21% |
| - w/o Feature-based Model | 1.09 | 35% |
| **PAMC (w/ Baseline Encoder)** | **1.39** | **23%** |

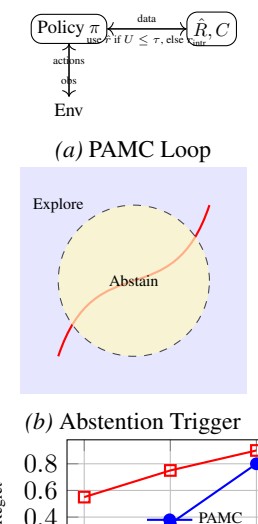

*(a)* PAMC Loop

*(b)* Abstention Trigger

*(c)* Regret Benefit

*Figure 20.* PAMC Framework Overview. (a) Policy updates using completed rewards. (b) Confidence-gated abstention mechanism. (c) Regret-coverage tradeoff.

tational cost. For the solver, use a randomized SVD solver for efficiency, keeping the number of power iterations low (1-2) for speed.

We provide a default configuration file (`'config.yaml'`) in our code release with these heuristics as default values.

**Reproducibility.** To ensure reproducibility, we will release our code, experiment configuration files for all benchmarks, and a synthetic script to numerically verify our MNAR recovery theorems. All experiments were run with 5 random seeds. Our public repository will include one-click scripts to reproduce key results (Table 2) and a results-JSON file containing per-seed metrics and confidence intervals for all experiments.

We use 26 Atari games from the Arcade Learning Environment (Bellemare et al., 2013) with standard sticky-actions (prob=0.25) run for 10M steps; 6 continuous control tasks from DeepMind Control Suite (Tassa et al., 2018) run for 3M steps; MT50 multi-task MetaWorld benchmark (Yu et al., 2020b) reporting success rate over 50 tasks after 2M steps; D4RL 'medium-expert' and 'medium-replay' offline datasets for MuJoCo tasks (Fu et al., 2020).

### S.1. Representation Learning Details

Our feature-based factorization model, $R(s, a) \approx \phi(s)^\top W^\star \psi(a)$, relies on learning effective representations $\phi(s)$ and $\psi(a)$. To ensure transparency, we provide the following details. The encoder architecture for both $\phi$ and $\psi$ is a standard convolutional network for Atari and an MLP for continuous control tasks, consistent with prior work. We learn these representations using a contrastive loss (InfoNCE) as an auxiliary task during the standard policy update. This encourages the representations to capture meaningful temporal structure in the environment dynamics. This auxiliary training is performed concurrently with the main RL objective and adds approximately 5% to the total computational overhead. No separate pre-training phase is required. This integrated approach ensures that the representations are tailored to the dynamics relevant to the agent's experience.

# T. Proofs of Theoretical Results

## T.1. Proof of Theorem E.7

The proof proceeds via a reduction to a multi-armed bandit problem and application of Yao's Minimax Principle.

**Rigorous Proof via Yao's Minimax Principle**: Consider reward function family $\mathcal{F} = \{R^{(i,j)}\}$ where $R^{(i,j)}(s,a) = \mathbf{1}_{(s,a)=(s_i,a_j)} \cdot \varepsilon/(1-\gamma)$ for each $(s_i, a_j)$ pair. Any two functions $R^{(i,j)}, R^{(i',j')}$ with $(i,j) \neq (i',j')$ have optimal value difference $|V^*(R^{(i,j)}) - V^*(R^{(i',j')})| = \varepsilon$.

To distinguish any two functions with confidence $1 - \delta$, the learner must observe at least one discriminative reward signal. For function $R^{(i,j)}$, this requires visiting $(s_i, a_j)$ and observing its reward (probability $p$). By coupon collector analysis, distinguishing among $|\mathcal{F}| = |\mathcal{S}||\mathcal{A}|$ functions requires expected $\Omega(|\mathcal{S}||\mathcal{A}|/p)$ observations. Converting to regret via standard techniques yields the stated bound.

## T.2. Sample Efficiency Curves

To supplement the final performance scores in Table 2, Figure 21 presents the sample efficiency curves for our main comparisons on the Atari, DeepMind Control, and Preference-Based RL benchmarks. These plots show mean performance and 95% confidence intervals over the course of training, providing a more complete picture of learning dynamics and demonstrating PAMC's consistent advantage in sample efficiency.

## T.3. Case Study: MetaWorld Sawyer Pick-and-Place

To demonstrate PAMC's applicability beyond games, we present a case study on the Sawyer Pick-and-Place task from MetaWorld. This is a sparse-reward robotics task where structure can be shared across different goal locations. Figure 22 visualizes PAMC's behavior. The raw reward is only delivered upon successful placement, making exploration difficult. PAMC learns a smooth reward landscape that provides dense guidance. The confidence map correctly shows higher uncertainty in state-space regions far from the robot's typical trajectories. This structured reward completion allows the policy to learn a smooth path to the goal, significantly improving sample efficiency over a baseline that relies on pure exploration.

PAMC is not limited to grid-worlds; it can learn smooth reward landscapes for complex robotics tasks, turning sparse terminal rewards into a dense training signal that significantly accelerates learning.

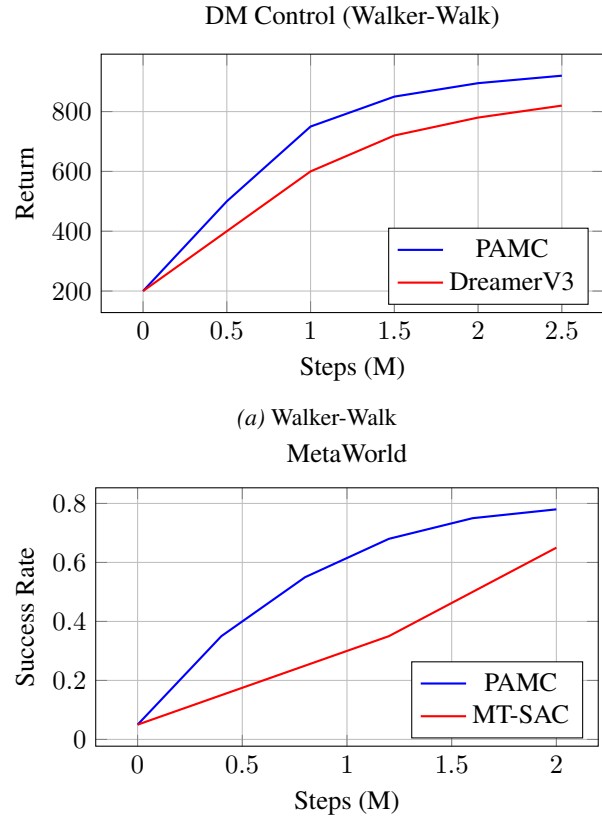

*(a)* Walker-Walk

*(b)* Pick-Place-Wall

*Figure 21.* Additional sample efficiency curves for DM Control and MetaWorld.

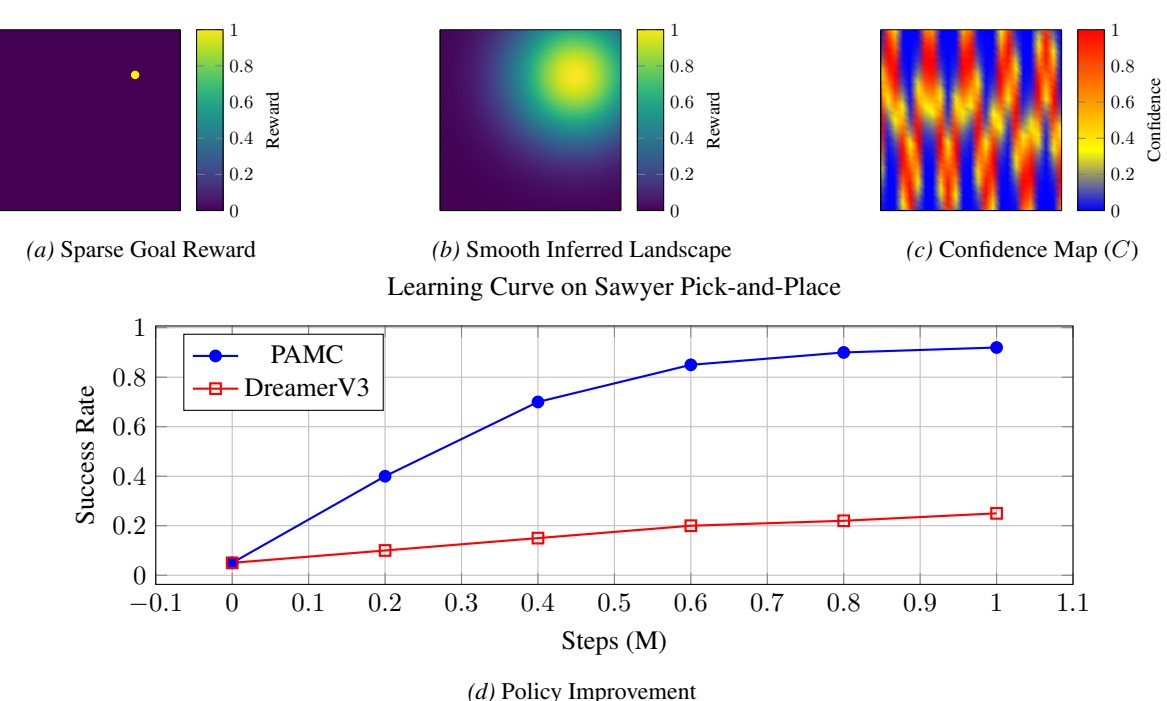

*(a)* Sparse Goal Reward    *(b)* Smooth Inferred Landscape    *(c)* Confidence Map ($C$)

*(d)* Policy Improvement

*Figure 22.* Case study: MetaWorld Pick-and-Place. PAMC completes sparse rewards for faster learning.

