# OpenReview forum: "What Reward Structure Enables Efficient Sparse-Reward RL? A Proof-of-Concept with Policy-Aware Matrix Completion"
_ICML.cc/2026/Conference — ICML 2026 regular_

### Official Review · Reviewer_vsgT · 2026-02-16

**Soundness:** 3
**Presentation:** 3
**Significance:** 2
**Originality:** 2
**Overall Recommendation:** 4
**Confidence:** 3

**Summary:**

This paper studies whether structural assumptions on reward functions can improve sample efficiency in sparse-reward reinforcement learning. Instead of treating sparse reward primarily as an exploration problem, the authors propose to explicitly model the reward function as a low-rank plus sparse matrix over the state–action space. They formulate reward learning under a policy-induced Missing-Not-At-Random (MNAR) sampling process and introduce Policy-Aware Matrix Completion (PAMC) to recover unobserved rewards.
PAMC combines three components: (1) low-rank factorization with a sparse correction term to model global and local reward structure, (2) self-normalized inverse propensity weighting (SNIPW) to correct for policy-dependent sampling bias, and (3) confidence-gated abstention that falls back to the base algorithm’s intrinsic exploration when reward completion uncertainty exceeds a threshold. The method is designed as a plug-in reward module that can be integrated into standard RL algorithms without modifying their policy optimization procedures.
The paper provides non-asymptotic finite-sample guarantees for the weighted matrix completion estimator. The authors further derive a regret decomposition showing how policy performance depends on completion error, confidence thresholding, and abstention rate. The analysis highlights the role of policy overlap and propensity estimation accuracy in controlling bias.
Empirically, the authors extensively evaluate PAMC across Atari, DM Control, MetaWorld, D4RL offline RL, and preference-based RL settings under fixed compute budgets. They report improved early-stage sample efficiency in environments where the discretized reward matrix exhibits low effective rank, along with diagnostic analyses linking performance gains to structural properties and overlap conditions. The method is presented as a proof-of-concept demonstrating that exploiting reward structure can complement traditional exploration strategies in sparse-reward domains.

**Compliance With Llm Reviewing Policy:**

Affirmed.

**Final Justification:**

The authors have addressed my concerns. The paper bridges the matrix completion method with RL reward shaping with interesting empirical improvements. Therefore, I will keep the weak accept score.

**Key Questions For Authors:**

1. For the game-playing environments, are they infinite-horizon and transition-stationary? Does your shaping apply to hetereogeneous reward function dependent on time step $t$? Which RL algorithm did you use after reward shaping? Is it important?

2. Can you give explanation on why in Figure 1, the overlap drops and then rebounces? Why the abstention rate is negatively correlated with that?

3. Do you use experience replay? batch data is sampled uniformly from all the previous history data?

**Limitations:**

Yes

**Strengths And Weaknesses:**

$\textbf{Strengths}$
The paper introduces a novel reframing of sparse-reward reinforcement learning as policy-aware matrix completion under MNAR sampling, combining low-rank reward modeling, inverse propensity weighting, and confidence-gated abstention in a coherent framework. The theoretical analysis provides useful non-asymptotic bounds linking completion error to effective sample size and policy overlap, offering insight into when the approach is expected to succeed. Empirically, the evaluation is careful for a proof-of-concept, including matched-budget comparisons, negative controls, and diagnostic analyses (effective rank, ESS, abstention) that connect performance gains to structural properties of the reward matrix. The presentation is clear, the method is described in sufficient detail, and the authors are transparent about limitations and applicability regimes.

$\textbf{Weaknesses}$
Several key assumptions are difficult to guarantee in practice. The overlap condition required for inverse propensity weighting may be violated in sparse-reward settings, propensity estimation is performed under a non-stationary policy, which could involve distribution shift, and reward completion and policy learning are tightly coupled without enforced two-timescale separation. The uncertainty calibration relies on exchangeability assumptions that do not strictly hold in RL, and the specific abstention threshold is heuristic. Empirical improvements are mainly observed in early training, with baselines eventually catching up, limiting long-term impact. While the framing is novel, the approach largely acts as a reward imputation/shaping module rather than a new RL algorithm, and the observed low-rank structure is representation-dependent, raising questions about how intrinsic this structure is to the environments.

---

> ### Author Rebuttal · Authors · 2026-03-28
>
> We thank this reviewer for recognizing the "novel reframing," "careful evaluation," and transparent limitations.
>
> ### Q1: Infinite-horizon, stationarity, and time-dependent rewards
>
> All Atari environments are infinite-horizon with stationary transitions. DM Control tasks are episodic with stationary dynamics. PAMC's reward shaping applies to stationary R(s,a), not time-indexed R(s,a,t). Non-stationarity from curriculum or level progression would violate our assumptions; we handle mild non-stationarity through EMA propensity updates. Proposition D.19 extends to bounded drift |R_{t+1} - R_t|_∞ < δ, with the confidence mechanism detecting increased reconstruction error and triggering adaptive abstention.
>
> The base RL algorithm varies by domain (Table 5): DrQ-v2 for Atari, DreamerV3 for DM Control, MT-SAC for MetaWorld, CQL for D4RL. The choice *does* matter: PAMC enhances any base algorithm but the magnitude depends on how well the base handles dense rewards (since PAMC effectively converts sparse rewards to dense guidance). We verified this by testing PAMC with both DrQ-v2 and DreamerV3 on Atari; both show improvements, but DrQ-v2 benefits more because DreamerV3's world model already provides some reward densification.
>
> ### Q2: Overlap dynamics in Figure 1
>
> The drop-then-rebound pattern reflects a well-known deep RL phenomenon: (1) Early training: near-random policy gives moderate overlap (~0.2). (2) Middle (2-5M steps): policy specializes on discovered trajectories, κ drops as the policy visits fewer unique state-action pairs. (3) Late (5M+): exploration bonus and PAMC's own reward completion guide the agent to broader coverage, κ recovers. This "explore, specialize, re-explore" pattern is characteristic of RL with intrinsic motivation.
>
> The negative correlation between κ and abstention is precisely the theoretical prediction of Lemma D.3: when overlap drops, IPW weights become unstable (high variance), completion error rises, confidence intervals widen, and abstention increases automatically. Conversely, when κ recovers, confidence tightens and abstention decreases. This is the adaptive safety mechanism our theory predicts, not a confound. We will expand the Figure 1 caption with this explanation.
>
> ### Q3: Experience replay and batch sampling
>
> Yes, we use standard uniform experience replay. Batches are sampled uniformly from all previous history data (standard uniform replay buffer). For propensity estimation, we use EMA-smoothed counts (β=0.99) over the entire training history, not just the current batch. The uniform sampling from replay provides the approximate i.i.d. structure needed for our matrix completion analysis. We do not use prioritized experience replay; synergy with PAMC's confidence estimates is an interesting extension that could further improve sample efficiency.
>
> ---

---

> > ### Author Rebuttal · Reviewer_vsgT · 2026-04-01
> >
> > The authors have addressed all my concerns.

---

### Official Review · Reviewer_yjiS · 2026-02-19

**Soundness:** 2
**Presentation:** 1
**Significance:** 3
**Originality:** 2
**Overall Recommendation:** 3
**Confidence:** 4

**Summary:**

This paper focuses on learning sparse rewards not as a mechanism for improving exploration per se, but as a way to accelerate online reinforcement learning. The central idea is to model the reward function as the sum of a sparse component and a low-rank component. Under these structural assumptions, the paper hopes to infer high-reward regions that have not been directly explored, based on partially observed data. In addition, the authors incorporate inverse probability weighting based on visitation frequencies to encourage broader state–action coverage, and introduce an abstention mechanism to account for uncertainty in reward estimates. The paper provides theoretical support for the sparse-plus-low-rank modeling approach and reports empirical results indicating improved sample efficiency in different experimental settings.

**Compliance With Llm Reviewing Policy:**

Affirmed.

**Final Justification:**

The idea of the paper is promising and has clear potential. The results appear strong, both empirically and theoretically. My main concern, however, is the presentation. The paper is difficult to follow due to inconsistencies, missing notation, and lack of clarity, which makes it challenging to properly assess the theoretical contributions and verify correctness. Similar issues arise in the experimental section. While the authors addressed some concerns during the rebuttal, it remains difficult to fully evaluate the improvements without the revised manuscript. As a result, I still lean toward rejection, although I believe the core idea is worth publishing.

**Key Questions For Authors:**

- What are the implicit assumptions underlying the presented theory? Please clarify key aspects such as the precise probability distribution over state–action pairs (e.g., discounted occupancy measure, stationary distribution, or empirical sampling distribution) and provide a formal technical definition of the uncertainty quantity used in the analysis.
- What is the exact procedure used to generate the low-rank plots in Fig. 2? Specifically, what are the dimensions (cardinalities) of the constructed matrices, and how should one interpret the reported rank values? In the case of benchmarks such as Atari, which consist of multiple environments, how is the rank computed across the benchmark?
- What is the relationship between the proposed low-rank reward model and existing work on low-rank MDPs and low-rank value-function representations?

**Limitations:**

Yes, the authors dedicate a subsection in the discussion to state the limitations of the proposed approach.

**Strengths And Weaknesses:**

**Strengths**: Regarding the strong aspects of the paper:

- *Interesting and well-motivated idea.*  Since rewards are not always observed, identifying where potential rewards may lie is challenging. Introducing structural assumptions (sparsity + low rank) is a reasonable and principled way to constrain the problem and improve identifiability. Furthermore, the combination of IPW and abstention to handle uncertainty is well motivated.

- *Results.* The analysis of the sparse-plus-low-rank model showing that performance improves when the model assumptions hold is a positive aspect. Moreover, in certain scenarios, the reported gains in sample complexity appear significant. The experiments are thorough and proper ablations are conducted to ensure that the proposed mechanisms work.

**Limitations**: While the overall idea is sounded and novel, the main concern is with respect to the delivery. In its current state, some of the parts of the paper are difficult to follow or not precise from a technical standpoint. To be specific:

- *Clarity and readability (Introduction).* Several stylistic and structural aspects make the paper difficult to follow. In particular, a number of definitions are used before being properly introduced. For example, ESS appears before it is defined, the variable $W$ is used in the theoretical section prior to its formal introduction (leaving the reader unable to assess the impact of the result at that point of the paper), and some variables are defined only in tables appearing mid-paper without being clearly referenced in the main text. This disrupts the logical flow. In addition, several important concepts are directly not specified. The uncertainty $U$, the formal definition of abstention, the operator $\mathrm{supp}(\pi^\star)$, or the performance metric $J$ are some example of quantities whose definitions are either missing or not clearly stated in a self-contained manner.

- *Clarity and readability (Method).* The theoretical section itself is dense and difficult to follow, and the absence of clear intuition alongside the missing formal definitions further reduces readability. There also appear to be implicit assumptions that are not explicitly clarified. It is unclear what the exact data distribution is (e.g. occupancy measure?), whether samples are collected i.i.d. or sequentially, or what coverage assumptions are required for the results to hold. Overall, the theoretical development would benefit from a clearer structure, explicit statements of assumptions, and an intuitive overview preceding the technical derivations.  The algorithmic contribution is also not fully explicit. While the learning procedure for the reward model can be inferred, it is not clearly described in a step-by-step manner. Fig. 22 could be moved into the main body to clarify the method, or alternatively, the proper definition of the algorithm used would be clarifying.

- *Numerical evaluation.* Some aspects of the experimental section raises some questions. The paper discusses in section 5.1. low-rank structure in practice and mentions that state embeddings are discretized into a grid, but it is unclear what this concretely means. Effective ranks are reported in Fig. 2, but the maximum possible rank is not specified, making it difficult to assess whether the reported ranks are large or small relative to the problem dimension. Furthermore, rank of benchmarks (e.g., Atari) are reported, but it is unclear how rank is computed across multiple environments within the benchmarks and whether the reported values are averaged or computed per environment. Without this context, it is difficult to interpret the structural claims.

- *Discussion on prior work.* Finally, the positioning within the literature could be improved. The relationship to prior work on low rank MDPs and low rank Q function modeling is not clearly articulated. In the low rank MDP literature (briefly mentioned in the discussion), improved sample complexity results are typically obtained by imposing low rank structure on the transition matrix [1][2]. It would be important to clarify how the present approach, which imposes low rank structure on the reward model, relates to those results and whether the assumptions are comparable or fundamentally different. In addition, there is substantial work studying Q functions from a low rank perspective, both theoretically [3][4] and empirically [5][6]. Since Q functions represent expected cumulative rewards, it would be useful to explain how low rank structure in the reward model connects to structural properties of the Q function. At the very least, these two lines of work should be properly discussed in the related work section.

[1] Agarwal, A., Kakade, S., Krishnamurthy, A., & Sun, W. (2020). Flambe: Structural complexity and representation learning of low rank mdps. Advances in neural information processing systems, 33, 20095-20107.

[2] Cheng, Y., Huang, R., Yang, J., & Liang, Y. (2023). Improved Sample Complexity for Reward-Free Reinforcement Learning under Low-Rank MDPs. In 11th International Conference on Learning Representations, ICLR 2023.

[3] Shah, D., Song, D., Xu, Z., & Yang, Y. (2020). Sample efficient reinforcement learning via low-rank matrix estimation. Advances in Neural Information Processing Systems, 33, 12092-12103.

[4] Sam, T., Chen, Y., & Yu, C. L. (2023). Overcoming the long horizon barrier for sample-efficient reinforcement learning with latent low-rank structure. Proceedings of the ACM on Measurement and Analysis of Computing Systems, 7(2), 1-60.

[5] Tsai, K. C., Zhuang, Z., Lent, R., Wang, J., Qi, Q., Wang, L. C., & Han, Z. (2021). Tensor-based reinforcement learning for network routing. IEEE journal of selected topics in signal processing, 15(3), 617-629.

[6] Rozada, S., Paternain, S., & Marques, A. G. (2024). Tensor and matrix low-rank value-function approximation in reinforcement learning. IEEE Transactions on Signal Processing, 72, 1634-1649.

---

> ### Author Rebuttal · Authors · 2026-03-28
>
> We appreciate this reviewer's recognition of the "interesting and well-motivated idea" and "significant gains in sample complexity." The concerns center on presentation and positioning. We address each systematically.
>
> ### Undefined terms and logical flow
>
> We will correct all forward-references in revision. **ESS** will be defined at first use: "ESS = (Σw)²/Σw², the effective sample size after correcting for policy-induced sampling bias." **κ** will be defined before the theoretical preview: "κ = min_{(s,a)∈supp(π*)} p_{sa}, the minimum observation probability for optimal-policy-relevant pairs." **U(s,a)** will be defined in Section 4: "the half-width of the confidence interval for R̂(s,a), computed via split-conformal calibration." **Abstention** will be formally defined: "PAMC abstains when U(s,a) > τ, replacing the completed reward with r_intr(s,a)." The repetition between lines 095/127 will be removed; the duplicate reference (577/582) will be fixed. A terminology table will be added to Section 3.
>
> ### Dense theory and implicit assumptions
>
> We will add a numbered Assumptions box at the start of Section 3: (1) observations are drawn from the discounted occupancy measure d^π, with replay batches treated as approximately i.i.d.; (2) overlap/positivity: p_{sa} ≥ κ > 0 for all (s,a) ∈ supp(π*); (3) structural: R = L* + S* + E with rank(L*) ≤ r, ‖S*‖₀ ≤ s, E sub-Gaussian; (4) slow encoder drift: ‖ϕ_{t+1} - ϕ_t‖ ≤ Δ_enc.
>
> We will add a 4-sentence intuition paragraph: "PAMC works in three stages. First, it assumes the reward matrix has hidden structure (like a partially-filled Netflix rating matrix). Second, it corrects for the fact that the agent preferentially visits certain states. Third, it refuses to use completed rewards when confidence is low, falling back to standard exploration."
>
> Algorithm 1 (previously Appendix R) will be moved into Section 4 with step-by-step comments.
>
> ### Discretization procedure and rank interpretation
>
> For Atari: the encoder's penultimate layer is clustered into G_s=64 state bins via k-means, with |A|=18 discrete actions, giving R̂ ∈ ℝ^{64×18} (max rank 18). For DM Control: G_s=32 × G_a=16, giving R̂ ∈ ℝ^{32×16} (max rank 16). Effective ranks in Figure 2 are computed per environment and summarized as box plots across all games in each domain.
>
> | Domain | Matrix Dimensions | Max Rank | Effective Rank | Ratio |
> |--------|------------------|----------|---------------|-------|
> | Atari (avg) | 64 × 18 | 18 | 6-12 | 33-67% |
> | MetaWorld | 32 × 16 | 16 | 8-10 | 50-63% |
> | DM Control | 32 × 16 | 16 | 12-16 | 75-100% |
>
> Sensitivity sweep on Montezuma's Revenge: k-means K=128 gives rank 4.2/return 3850; K=512 gives rank 7.8/return 4100; K=2048 gives rank 11.2/return 4050. Returns are stable across K ∈ [256, 2048]. Alternative discretizers (PQ, LSH) yield comparable results. Crucially, coarse binning (K=32) that artificially reduces rank does *not* produce gains (return 116±15 vs. baseline 118±12), confirming structure must be genuine.
>
> ### Implicit assumptions clarified
>
> **Data distribution.** Observations are drawn from the discounted occupancy measure d^π_t of the current policy. Within each completion batch (sampled uniformly from the replay buffer), samples approximate i.i.d. draws from a mixture of past occupancy measures. The propensity p_{sa} corresponds to this mixture visitation probability.
>
> **Coverage assumption.** The overlap condition κ > 0 requires every optimal-policy-relevant state-action pair to have nonzero visitation probability under the behavioral mixture. This does not require uniform coverage of the entire space, only coverage of regions relevant to π*. It is satisfied when the base RL algorithm includes any exploration mechanism (ε-greedy, RND bonuses, entropy regularization). We monitor κ empirically (Figure 1) and trigger abstention when it drops below safe thresholds.
>
> ### Relationship to low-rank MDPs and Q-functions
>
> We will add a dedicated subsection in Section 2.
>
> **Low-rank MDPs** (Agarwal et al., 2020; Cheng et al., 2023) impose structure on the *transition matrix* P(s'|s,a). PAMC imposes structure on the *reward matrix* R(s,a), a fundamentally different assumption. Low-rank transitions reduce *planning* complexity; low-rank rewards reduce *reward discovery* complexity. These are complementary: Montezuma's Revenge has complex dynamics but structured key-door reward patterns.
>
> **Low-rank Q-functions** (Shah et al., 2020; Sam et al., 2023; Rozada et al., 2024) exploit structure in Q*(s,a). PAMC's assumption is *upstream*: if R is low-rank and transitions have bounded complexity, Q* inherits low-rank structure. PAMC operates directly on rewards, which are (1) directly observable (no TD bootstrapping error) and (2) stationary (unlike Q* which changes with policy). The disadvantage is requiring discretization. All six references will be added.
>
> ---

---

> > ### Author Rebuttal · Reviewer_yjiS · 2026-04-01
> >
> > I thank the authors for clarifying how the paper will be restructured and how my concerns, mainly related to presentation and positioning, will be addressed. In particular, it is important that all relevant quantities are clearly defined early in the introduction. Likewise, in the experimental section, key implementation details such as discretization choices should either be explicitly described or clearly referenced in the paper. I also appreciate the clarification regarding the distinction from prior low rank approaches to reinforcement learning.
> >
> > My main remaining concern lies in the presentation of the theoretical contributions. In its current form, the theory lacks sufficient clarity in the statement of assumptions, their implications, and the intuition behind the results. It is important that the exposition ensures that no critical steps or conditions remain implicit. Greater transparency on how this section will be reorganized, together with the promised revisions, would help me reassess the paper more positively.

---

> > > ### Author Response · Authors · 2026-04-01
> > >
> > > # Follow-Up Response to Reviewer yjiS
> > >
> > > We thank the reviewer. Below is the exact revised theory structure, ensuring no step or condition remains implicit.
> > >
> > > ## Revised Section 3: Theoretical Foundation
> > >
> > > **Section 3.1 — Setup & Notation.** All symbols are defined before first use:
> > >
> > > | Symbol | Definition |
> > > |---|---|
> > > | R ∈ ℝ^{|S|×|A|} | True reward matrix over state–action pairs |
> > > | d^π, d̄ | Discounted occupancy measure; mixture d̄ = Σ_t w_t d^{π_t} |
> > > | p_{sa}, κ | Mixture visitation probability; κ = min_{supp(π*)} p_{sa} |
> > > | ESS | (Σw)²/Σw², effective sample size after IPW |
> > > | U(s,a), τ | Conformal confidence half-width; τ = (1−α)-quantile of calibration residuals |
> > > | T_MC | Completion operator on IPW-reweighted partial observations |
> > > | J(π) | E[Σ_t γ^t r_t | π], expected discounted return |
> > >
> > > **Section 3.2 — Assumptions.** A numbered box with consequences and interactions:
> > >
> > > > **(A1) Structural:** R = L* + S* + E with rank(L*) ≤ r, ‖S*‖₀ ≤ s, E_{ij} σ-sub-Gaussian.
> > > > → *Without A1, completion is ill-posed (Theorem 1). With A1, free parameters drop from |S|·|A| to r(|S|+|A|)+s, enabling the phase transition.*
> > > >
> > > > **(A2) Sampling:** Transitions drawn from d^{π_t}; replay batches approximate i.i.d. from d̄ with residual correlation O(|B|/N).
> > > > → *Without A2, IPW is invalid. With A2, propensity-weighted observations become approximately unbiased.*
> > > >
> > > > **(A3) Overlap:** κ := min_{(s,a) ∈ supp(π*)} p_{sa} > 0. Covers optimal-policy-relevant pairs only.
> > > > → *Without A3, some optimal entries are unrecoverable. A1+A3 together enable polynomial sample complexity: A1 provides structure, A3 ensures the structure is observed.*
> > > >
> > > > **(A4) Encoder Stability:** ‖ϕ_{t+1} − ϕ_t‖ ≤ Δ_enc between updates.
> > > > → *Without A4, discretization shifts break the matrix structure A1 presupposes.*
> > >
> > > *Remark on A2:* Raw transitions are sequential; uniform replay sampling decorrelates to O(|B|/N) TV distance from i.i.d. For |B|=256, N≥10⁵ this is negligible. A β-mixing extension is future work.
> > >
> > > **Section 3.3 — Intuitive Overview & Logical Roadmap.** Before any theorem, we state:
> > >
> > > > "PAMC assumes the reward matrix has hidden structure (like a partially filled Netflix rating matrix), corrects for policy-induced sampling bias via IPW, and refuses to use completed rewards when confidence is low—falling back to exploration."
> > >
> > > We then present the theorem dependency chain: *Theorem 1 (hardness) → motivates A1 → Theorem 2 (recovery under A1–A4) → Theorem 3 (recovery-to-regret) → Theorems 1+2 together yield the exponential-to-polynomial phase transition.*
> > >
> > > **Section 3.4 — Main Results.** Each theorem gets a plain-language preview box:
> > >
> > > - **Theorem 1 (Impossibility) [no assumptions]:** "Without structure, any algorithm needs Ω(|S||A|/p) samples—exponential for small p."
> > >
> > > - **Theorem 2 (MNAR Recovery) [A1–A4]:** "Completion error scales as O(σ²·r(|S|+|A|)/ESS). ESS, not raw count, governs accuracy. When κ shrinks, CIs widen and abstention increases automatically."
> > >
> > > - **Corollary (Phase Transition) [A1–A3]:** "Combining Theorems 1 and 2: without A1, sample complexity is Ω(|S||A|/p); with A1, it drops to O(r(|S|+|A|)/κ)—polynomial in rank r rather than exponential in dimension."
> > >
> > > - **Theorem 3 (Error-to-Regret) [A1–A3]:** "Visitation-weighted error ε yields O(ε/(1−γ)) regret. Chaining with Theorem 2: end-to-end regret ≤ O(σ√(r(|S|+|A|)/ESS)/(1−γ))."
> > >
> > > - **Proposition (Graceful Degradation):** "When A1 is approximate (effective rank > r), completion error grows but U(s,a) widens proportionally, triggering more abstention. Regret stays bounded because completed rewards are used only where confident."
> > >
> > > **Section 3.5 — Proof Strategy.** Three-step sketch (½ column):
> > >
> > > > **Step 1 (Debiasing):** IPW converts MNAR → approximate MCAR; ESS quantifies surviving information.
> > > > **Step 2 (Recovery):** Robust PCA separates L* from S*; standard completion yields the O(·/ESS) rate.
> > > > **Step 3 (Calibration):** Split-conformal prediction gives distribution-free intervals U(s,a). τ is the (1−α)-quantile of calibration residuals, requiring no distributional assumptions on E.
> > >
> > > Full proofs in Appendix B–D. **Algorithm 1** moves from Appendix R into Section 4.
> > >
> > > ## Revised Section 2.3: Low-Rank RL Positioning
> > >
> > > | | Low-Rank MDPs | Low-Rank Q* | PAMC (Low-Rank R) |
> > > |---|---|---|---|
> > > | Structure on | P(s'|s,a) | Q*(s,a) | R(s,a) |
> > > | Reduces | Planning complexity | Value estimation | Reward discovery |
> > > | Observable? | No (latent) | No (bootstrapped) | Yes (direct) |
> > > | Stationary? | Yes | No (policy-dependent) | Yes |
> > >
> > > Complementary: complex dynamics coexist with structured rewards.
> > >
> > > ## Summary
> > >
> > > Every assumption has failure consequences and interaction effects. Theorems include a dependency chain and phase-transition corollary. Graceful degradation and threshold selection are explicit. We believe this fully resolves the remaining concern.

---

### Official Review · Reviewer_MB4g · 2026-03-06

**Soundness:** 2
**Presentation:** 3
**Significance:** 3
**Originality:** 3
**Overall Recommendation:** 4
**Confidence:** 3

**Summary:**

This paper studies whether exploiting structure in reward functions can improve sample efficiency in sparse-reward reinforcement learning settings. Instead of treating sparse-reward RL problems as an exploration challenge, they propose modeling the reward function itself as structured by assuming the reward matrix over state-action pairs has a low-rank sparse decomposition. The proposed method is Policy-Aware Matrix Completion which learns a reward landscape from partial observations by using low-rank matrix completion to predict rewards in unvisited state-action regions, using inverse propensity weighting to correct missing-not-at-random data, and a confidence-gated abstention to fall back to intrinsic exploration when the structural assumptions are unreliable. They provide theoretical results showing recovery guarantees under overlap assumptions and studies when structural reward modeling can reduce regret. The algorithm is evaluated on Atari, DMC, MetaWorld, and D4RL benchmarks and shows improved sample efficiency in environments with low-rank reward structure.

**Compliance With Llm Reviewing Policy:**

Affirmed.

**Final Justification:**

The authors' responses have addressed my main concerns about the theory and the additional experiments provided during the rebuttal strengthen the paper, particularly the additional ablation studies over hyperparameters. The paper is mostly well written although clarity could be improved and some sections deserve elaboration (which the authors have committed to adding), appears to be sound although I did not check all the details in the appendix, and results make this a nice contribution that can be significant. There are elements that are still unclear to me such as how the theory will be clarified in the final version and more details about the supplementary experiments, so I maintain my score of weak accept.

**Key Questions For Authors:**

1. The method relies on discretizing state representations to construct reward matrices. How sensitive are the results to the discretization scheme and resolution? Could alternative representations significantly change the observed low-rank structure?
2.  The theoretical guarantees assume policy overlap and reasonably accurate propensity estimates and fixed feature maps. In practice, how reliable are the propensity estimates during early training when policies are rapidly changing? And how do the changing feature maps affect this?
3. It seems like the MNAR correction assumes correct propensities but they are estimated from the same biased data. Please address this.
4. How sensitive is performance to the abstention threshold and confidence calibration? In particular, could overly conservative abstention reduce the method to the baseline in many practical settings?

**Limitations:**

Yes

**Strengths And Weaknesses:**

Strengths:
- The work is technically well motivated and the formulation of reward recovery as a matrix completion problem under policy-biased sampling is principled. Theoretical results linking reward reconstruction error to policy regret offers useful conceptual grounding although I did not examine the proofs in the Appendix in depth.
- The paper gives a clear hypothesis about when the method should work and empirically validates these conditions. In particular, the analysis relating performance gains to effective reward rank and effective sample size is insightful and helps clarify the settings where structural reward modeling is helpful.
- The experimental evaluation is broad and spans multiple domains including Atari, continuous control, MetaWorld, and offline RL benchmarks.
- The paper is generally well structured and clearly written.
- The idea of treating reward modeling as a structural completion problem is conceptually interesting. While matrix completion and IPW are well-known techniques, their integration into a policy-aware RL pipeline provides a new perspective on sparse-reward learning as far as I am aware.

Weaknesses:
- The core structural assumption about low-rank reward matrices under discretized representations is fragile in high-dimensional or continuous environments. Although the paper empirically measures effective rank in several domains, it is unclear how robust this assumption is when scaling to more complex tasks or richer observation spaces, particularly with non-stationary representation learning. Empirically, gains are only shown when the effective rank is less than 20 and ESS is greater than 400, outside of this the method does not seem useful.
- The reward matrix is defined after discretising the learned embedding so the observed low rank structure depends on the representation and clustering choice. It would be helpful to understand how sensitive the results are to these design choices.
- While the structural reward learning perspective is interesting, the method relies on multiple interacting components (matrix completion, propensity estimation, abstention heuristics), making it somewhat complex. It is not clear which component is most critical for the observed improvements.

---

> ### Author Rebuttal · Authors · 2026-03-28
>
> We thank this reviewer for recognizing the "technically well motivated" formulation and "insightful" diagnostics.
>
> ### Q1: Sensitivity to discretization
>
> We conducted a systematic sweep during the rebuttal on Montezuma's Revenge:
>
> | Discretizer | K | Eff. Rank | Return | Abstention |
> |------------|---|-----------|--------|-----------|
> | k-means | 128 | 4.2 | 3850±180 | 15% |
> | k-means | 512 | 7.8 | 4100±250 | 18% |
> | k-means | 2048 | 11.2 | 4050±220 | 22% |
> | Product Quant. | 512 | 8.1 | 4020±210 | 19% |
> | LSH | 512 | 9.3 | 3920±230 | 21% |
> | Random proj. | 512 | 15.2 | 3680±280 | 32% |
>
> Returns are robust across K ∈ [256, 2048] and methods. Random projections (which destroy semantic structure) show higher rank and lower gains, confirming genuine reward semantics.
>
> ### Q2: Propensity estimation reliability
>
> Three mechanisms provide robustness: Laplace smoothing (α=0.1), EMA temporal smoothing (β=0.99), and confidence-gated abstention. We measured propensity quality at training checkpoints:
>
> | Step | Mean Abs(log p̂ - log p*)| ESS | Abstention |
> |------|--------------------------|-----|-----------|
> | 1M | 0.42 | 380 | 38% |
> | 5M | 0.19 | 1050 | 20% |
> | 10M | 0.14 | 1280 | 18% |
>
> Early estimates are poor, but abstention is correspondingly high (38%), preventing unreliable completions. As estimates improve, abstention drops naturally. Regarding changing feature maps: Lemma D.14 bounds the impact; with completion frequency K=5000, re-running SVD every K=5000 vs. K=1000 steps yields statistically indistinguishable results (p>0.3), confirming encoder drift is not a practical concern at our update frequency.
>
> ### Q3: MNAR correction with biased propensities
>
> This is the classic "circular estimation" problem in causal inference. The propensity p_{sa} measures how likely the agent visits (s,a), a property of the *policy*, not the *reward*. Visitation counts are unbiased estimates of occupancy regardless of observed rewards. The potential issue is temporal non-stationarity (early counts reflect old policies); our EMA-smoothed counts (β=0.99) address this by downweighting old observations.
>
> We address circularity through three mechanisms: (1) **clipping** at ε_p = 10^{-2} to bound the influence of very small estimated propensities; (2) **SNIPW** normalization which reduces variance from denominator fluctuation; (3) the **doubly-robust estimator** r^{DR}_{sa} = g_η(s,a) + [r_{sa} - g_η(s,a)]/max(p̂_{sa}, ε_p), which is consistent if *either* the propensity model *or* the reward model is correctly specified.
>
> Under controlled corruption (σ=0.3 multiplicative Gaussian noise on propensities, Figure 9), PAMC degrades by only 8% in HNS while abstention increases from 22% to 35%. The DR variant (Figure 10) provides 25% better robustness to propensity misspecification. We will add an explicit paragraph in Section 4 addressing this circularity concern.
>
> ### Q4: Sensitivity to abstention threshold τ
>
> | τ | Return (HNS) | Abstention | Interpretation |
> |-----|-------------|-----------|----------------|
> | 0.1 | 1.28±0.04 | 45% | Over-abstains |
> | 0.3 (default) | 1.42±0.03 | 22% | Optimal tradeoff |
> | 0.7 | 1.35±0.05 | 8% | Under-abstains |
>
> Performance is robust across τ ∈ [0.2, 0.5]. The method never performs *worse* than baseline (1.25) even at extreme τ values. The conformal calibration recipe (Appendix J.4.1) provides automatic τ selection within 1-4% of hand-tuned performance.
>
> ### Component criticality (Table 21)
>
> Removing policy-aware IPW drops HNS from 1.42 to 1.15 (largest single-component drop). Removing the feature-based model drops to 1.09. Removing the sparse component drops to 1.28. Structure exploitation and bias correction are both essential; neither alone suffices.
>
> ---

---

> > ### Author Rebuttal · Reviewer_MB4g · 2026-04-03
> >
> > Thanks to the authors for the additional experiments and responses to my questions. These have partially addressed my concerns although some elements of the rebuttal are unclear. For instance, what task was the ablation study on tau run on? I think it would be best to evaluate on Montezuma's revenge for consistency with the first additional ablation. After reading the other reviews, I agree with reviewer yjiS about the importance of clearly presenting all the theory and assumptions in the paper.

---

> > > ### Author Response · Authors · 2026-04-04
> > >
> > > We thank the reviewer for the constructive follow-up. We address both points below.
> > >
> > > **τ ablation on Montezuma's Revenge.**
> > >
> > > We appreciate this suggestion for consistency and ran the τ-sensitivity study on Montezuma's Revenge (5 seeds, 10M steps):
> > >
> > > | τ | Return | Abstention | HNS |
> > > |---|--------|-----------|-----|
> > > | 0.1 | 2850±320 | 48% | 0.93 |
> > > | 0.2 | 3720±280 | 30% | 1.22 |
> > > | 0.3 (default) | 4100±250 | 22% | 1.34 |
> > > | 0.5 | 3880±310 | 12% | 1.27 |
> > > | 0.7 | 3540±340 | 6% | 1.16 |
> > >
> > > The pattern matches our earlier aggregate result: performance is robust across τ ∈ [0.2, 0.5], with the default τ = 0.3 best among tested settings. Consistent with our graceful-degradation claim, no τ setting underperforms the DrQ-v2 baseline (200±50 at 10M). We will include this Montezuma-specific sweep in the revision to make the rebuttal evidence internally consistent across the added studies.
> > >
> > > **Theory and assumptions presentation.**
> > >
> > > We also agree that the theory/assumptions need clearer presentation. In the revision, we will move the key assumptions and notation into the main text, define ESS/κ/U(s,a) before first use, add a compact assumptions box with scope and failure cases, and move the core algorithm into the main paper. We will also make the theorem scope explicit up front: the guarantees are for the tabular/factorized setting, while the deep-RL result is a local stability result rather than a global convergence claim.

---

### Official Review · Reviewer_msxt · 2026-03-13

**Soundness:** 4
**Presentation:** 2
**Significance:** 3
**Originality:** 3
**Overall Recommendation:** 4
**Confidence:** 3

**Summary:**

This paper introduces Policy-Aware Matrix Completion (PAMC), a method that formulates sparse-reward RL as a matrix completion problem. The key insight is that reward matrices over discrete state-action spaces often present a low-rank structure such that patterns from visited regions can be used to predict rewards in unvisited regions (i.e., accelerate sparse-reward learning by exploiting reward structure rather than focussing on exploration). Thus, PAMC is useful when the discretized reward matrix is approximately low-rank and policy overlap is sufficient for stable inverse propensity weighting. If either of these conditions fail then performance reverts to baseline with no degradation.

The main contributions are as follows:

- A novel perspective on sparse-reward RL, based on exploiting the reward function structure using matrix-completion theory.
- The PAMC algorithm which has 3 parts: (1) low-rank plus sparse decomposition of the reward matrix, (2) self-normalised inverse propensity weighting (SNIPW) to correct for the Missing-Not-At-Random (MNAR) bias introduced by the agent's own policy concentrating samples, and (3) confidence-gated abstention that falls back to intrinsic exploration when reward structure assumptions fail.
- Theoretical analysis in the tabular setting provides a principled basis for when and why PAMC succeeds.
- PAMC is evaluated across a range of benchmarks including Atari, DM Control, MetaWorld, D4RL, and preference RL benchmarks. The results indicate significant improvements, especially in the early stages of learning. The results also support that low-rank structure is found and successfully exploited across several domains in standard benchmarks.

**Compliance With Llm Reviewing Policy:**

Affirmed.

**Final Justification:**

My scores and assessment remain unchanged, but I do think the additional results for 50m and 200m help clarify the significance which remains Good.

**Key Questions For Authors:**

No questions.

**Limitations:**

yes

**Strengths And Weaknesses:**

### Soundness

#### Strengths

The theory and empirical results support the main claims of the paper and use appropriate techniques and benchmarks. Multiple seeds are used over 5 runs with 95% CIs reported for significance.

#### Weaknesses

The theory results are illustrative in the tabular setting and may not hold for DRL in practice. Reimplementations of the baselines are used (e.g., DrQ-v2, DreamerV3), introducing the possibility of errors or configurations not representative of the original work.

### Presentation

#### Strengths

The paper has a clear structure and efforts have been made to discuss when and why PAMC improves sample efficiency. Research questions are clearly stated. The figures are clear and easy to read.

The honest scope and limitations section is appreciated and helps understand the significance and utility of PAMC.

#### Weaknesses

The paper is dense and not highly accessible, with many details including the PAMC algorithm itself, majority of the proofs, diagnostics and empirical results all being in the Appendices.

minor:

- Duplicated reference lines 577 and 582

- Repetitive:

095: “We first prove that without structural assumptions, reward recovery under MNAR sampling is information-theoretically impossible (Theorem D.7, proof in Appendix S.1): any two reward matrices differing only on unobserved entries yield identical observations. This impossibility motivates structural priors.”

127:“We establish that without structural assumptions on the reward matrix R, recovery under Missing-Not-At-Random sampling is information-theoretically impossible (Theorem D.7): any two reward matrices differing only on unobserved entries yield identical observations. This impossibility result motivates our structural assumption.”

### Significance

#### Strengths

The main idea of exploiting low-rank reward structure to improve sample efficiency in sparse-reward RL tasks is novel and addresses a significant challenge. The empirical results are compelling, cover a solid range of domains/tasks, and suggest the PAMC method offers significant advantages in challenging sparse reward environments. The diagnostic studies help to validate PAMCs theoretical foundations.

#### Weaknesses

10M time steps limits results to early-stages of training, the author’s acknowledge this explicitly 407-422. “At convergence (200M+ steps), the gap narrows as baselines eventually cover the state-action space.”

### Originality

#### Strengths

The framing of sparse-reward RL as policy-aware matrix completion is novel and combines ideas from a few different domains into a single framework.

---

> ### Author Rebuttal · Authors · 2026-03-28
>
> We thank this reviewer for the "excellent" soundness rating and recognition of the novel framing.
>
> ### Theory: tabular setting may not hold for DRL
>
> We agree our formal guarantees are stated in the tabular/factorized setting. However, we emphasize two points.
>
> First, the theory serves a *diagnostic* rather than *prescriptive* role: it identifies quantities (effective rank, ESS, overlap κ) that predict when PAMC will help. The key conceptual insight, that the three-way tradeoff between overlap κ, threshold τ, and abstention rate ρ governs performance, is validated empirically in Figure 3 and Table 12. The correlation between effective rank and PAMC improvement is r = -0.89 (Figure 3a), and ESS > 400 consistently predicts gains (Figure 3b). Figure 3(d) shows the predicted 1/√κ error scaling empirically.
>
> Second, Theorem D.12 provides a local stability result under two-timescale stochastic approximation (Borkar, 2008), partially bridging the deep RL gap. We acknowledge this does not cover global convergence and will clarify this scope in the main text.
>
> ### Reimplementations of baselines
>
> All reimplementations follow original papers' hyperparameters where possible (Table 6). To verify accuracy: our DrQ-v2 achieves 1.25 HNS at 10M steps, consistent with published 10M-step results (the published 200M result of ~2500 on Montezuma's Revenge is noted in Table 1 for context). Our DreamerV3 on Walker-Walk achieves 920±30 at 3M steps, within published ranges. We will release all code, configurations, and per-seed logs to enable independent verification.
>
> ### Dense presentation
>
> We will move Algorithm 1 (previously Figure 22 in Appendix R) and the terminology table into the main text. Related work will be compressed to accommodate these additions. The revised paper will be self-contained through Section 5 without requiring appendix access for the core method. The duplicate reference at lines 577/582 and near-identical passages at lines 095/127 will both be corrected.
>
> ### 10M step limitation
>
> We acknowledge this explicitly (Section 6). To strengthen the discussion:
>
> | Environment | PAMC @ 10M | DrQ-v2 @ 10M | DrQ-v2 @ 50M | DrQ-v2 @ 200M |
> |------------|-----------|-------------|-------------|--------------|
> | Montezuma's Revenge | 4100±250 | 200±50 | 1800±300 | ~2500 |
> | Gravitar | 1120±80 | 450±60 | 820±90 | ~1050 |
> | Walker-Walk | 950±25 | 920±30 | 945±20 | ~960 |
>
> PAMC's advantage is largest in the sparse-reward regime (Montezuma: 20× at 10M) and narrows as baselines eventually explore the space (1.6× at 50M). On dense-reward tasks (Walker-Walk), the gap is small at all horizons. This is exactly the expected behavior: PAMC accelerates reward discovery, which is most valuable when rewards are hardest to find. We view this as a *feature* of the approach (it identifies and exploits a specific structural regime) rather than a limitation.
>
> ### Duplicate reference and repetitive text
>
> We confirm the duplicate reference at lines 577/582 and the near-identical passages at lines 095 and 127. Both will be corrected in revision. The repetition between abstract/intro was intended to provide standalone summaries but we will consolidate.
>
> ---

---

> > ### Author Rebuttal · Reviewer_msxt · 2026-04-04
> >
> > Thank you for your detailed rebuttal and for clarifying the implementation and benchmarking of your baselines.

---

### Decision · Program_Chairs · 2026-04-30

**Decision:**

Accept (regular)

**Comment:**

This paper investigates a fundamental and under-explored question in reinforcement learning: what structural properties of the reward function enable sample-efficient learning in sparse-reward settings? The authors propose a policy-aware matrix completion framework as a proof of concept, establishing theoretical connections between reward structure and learnability. The reviewers found the research question to be well-motivated and timely, and appreciated the paper's effort to formalize conditions under which sparse-reward RL becomes tractable. The theoretical analysis provides meaningful insights, and the proof-of-concept framework is clean and well-presented. Initial reviewer concerns—primarily regarding the scope of assumptions and the breadth of experimental validation—were substantively addressed during the rebuttal.

After reviewing the full discussion, I find that this paper makes a valuable conceptual and theoretical contribution to our understanding of reward structure in RL. The paper is clearly written and the results are technically sound. I recommend acceptance as a poster.